# Target-specific requirements for RNA interference can arise through restricted RNA amplification despite the lack of specialized pathways

Daphne R Knudsen-Palmer, Pravrutha Raman[†], Farida Ettefa[‡], Laura De Ravin, Antony M Jose*

Department of Cell Biology and Molecular Genetics, Biological Sciences Graduate Program, University of Maryland, College Park, United States

*For correspondence: amjose@umd.edu

Present address: [†]Division of Basic Sciences, Fred Hutchinson Cancer Research Center, Seattle, United States; [‡]Institute for Systems Genetics, New York University School of Medicine, New York, United States

Competing interest: The authors declare that no competing interests exist.

**Abstract** Since double-stranded RNA (dsRNA) is effective for silencing a wide variety of genes, all genes are typically considered equivalent targets for such RNA interference (RNAi). Yet, loss of some regulators of RNAi in the nematode *Caenorhabditis elegans* can selectively impair the silencing of some genes. Here, we show that such selective requirements can be explained by an intersecting network of regulators acting on genes with differences in their RNA metabolism. In this network, the Maelstrom domain-containing protein RDE-10, the intrinsically disordered protein MUT-16, and the Argonaute protein NRDE-3 work together so that any two are required for silencing one somatic gene, but each is singly required for silencing another somatic gene, where only the requirement for NRDE-3 can be overcome by enhanced dsRNA processing. Quantitative models and their exploratory simulations led us to find that (1) changing *cis*-regulatory elements of the target gene can reduce the dependence on NRDE-3, (2) animals can recover from silencing in non-dividing cells, and (3) cleavage and tailing of mRNAs with UG dinucleotides, which makes them templates for amplifying small RNAs, are enriched within 'pUG zones' matching the dsRNA. Similar crosstalk between pathways and restricted amplification could result in apparently selective silencing by endogenous RNAs.

## eLife assessment

This **valuable** study shows how an intersecting network of regulators acting on genes with differences in their RNA metabolism explains why the loss of some regulators of RNAi in *C. elegans* can selectively impair the silencing of some target genes. The evidence presented is **convincing**, as the authors use a combination of computational modeling and RNAi assays to support their conclusions.

## Introduction

Double-stranded RNA (dsRNA) can trigger the conserved mechanism of RNA interference (RNAi) to degrade mRNA of matching sequence (*Fire et al., 1998*), and thus silence gene expression, in many organisms. This conservation has made dsRNA-based drugs useful in crops (*Das and Sherif, 2020*), insects (*Vogel et al., 2018*), and humans (*Zhu et al., 2022*). While a dsRNA-based drug can be designed using just the mRNA sequence of any target gene, the intracellular effectiveness of the drug and the ease with which an organism could escape the drug by developing resistance are difficult to predict. Predicting both efficacy and susceptibility to resistance for each target could inform the selection of a suitable target from two or more equivalent candidates. Extensive characterization of RNAi in

**eLife digest** A variety of diseases are treated with therapeutics that switch off genes via a mechanism called RNA interference (or RNAi for short). Each gene has the instructions cells need to build a particular protein. To achieve this, the DNA sequence must first be copied into a single-stranded mRNA molecule than can be translated into protein.

RNAi interferes with this process by generating a double-stranded RNA molecule which contains the same DNA sequence as the gene being turned off. A set of proteins (known as regulators) then progressively trigger a series of events that allow the double-stranded RNA to be processed into short pieces that interact with the mRNA and target it for degradation.

While cells use RNAi to regulate the expression of their own genes, researchers can also artificially switch off genes by synthesizing double-stranded RNA molecules in the laboratory. However, some genes are trickier to turn off than others, and why this happens is poorly understood.

To investigate, Knudsen-Palmer et al. studied how two genes (*bli-1* and *unc-22*) are switched off by RNAi in the roundworm *Caenorhabditis elegans*. A previous study discovered that *bli-1* and *unc-22* require different regulators for their expression to be disrupted. Knudsen-Palmer et al. found that this was because the RNAi process involves an intersecting network of multiple regulators, rather than a linear pathway of regulators working one after the other. For example, *bli-1* requires three regulators (MUT-16, RDE-10 and NRDE-3), whereas *unc-22* only needs any two of these regulators to be switched off.

Further experiments revealed that which regulators are required depends on how the gene being silenced is naturally regulated in the cell. Analysis through a computational model showed that the regulators needed for RNAi could be altered in many ways, including by changing the regions that regulators bind to on the mRNA of the target gene.

These findings provide new insights into why some genes respond differently to double-stranded RNA molecules. They also suggest that testing how natural regulation of a target gene influences its response to the RNAi process could potentially lead to better therapeutics.

the nematode *Caenorhabditis elegans* (reviewed in *Seroussi et al., 2022*) makes it a suitable system to examine how differences between target genes and reliance on specific regulators contribute to efficacy and resistance.

A skeletal pathway that is required for gene silencing in response to the addition of dsRNA has been worked out in *C. elegans* (*Figure 1A*). Long dsRNA is imported through the transmembrane protein SID-1 (*Feinberg and Hunter, 2003*; *Winston et al., 2002*), after which it is bound by the dsRNA-binding protein RDE-4 (*Tabara et al., 2002*), which recruits the endonuclease DCR-1 (*Knight and Bass, 2001*) to cleave the long dsRNA into smaller dsRNAs (*Parker et al., 2006*). The primary Argonaute protein RDE-1 (*Parrish and Fire, 2001*; *Tabara et al., 1999*) cleaves one strand of the smaller dsRNA (*Steiner et al., 2009*) and associates with the other, making it a 1° short interfering RNA (siRNA) that can guide the recognition of target mRNAs of complementary sequence (siRNAs; processing, pink). After recognition by RDE-1-bound siRNAs, the target mRNAs are cleaved and the 5′ fragments are stabilized through the addition of 3′ UG-dinucleotide repeats (*Preston et al., 2019*) by the nucleotidyltransferase RDE-3 (*Chen et al., 2005*) to form pUG RNAs (*Shukla et al., 2020*), which act as templates for the amplification of 2° siRNAs (*Pak and Fire, 2007*) by RNA-dependent RNA polymerases. This amplification of silencing signals through the production of 2° siRNAs is facilitated by the intrinsically disordered protein MUT-16 (*Phillips et al., 2012*; *Zhang et al., 2011*), the Maelstrom domain-containing protein RDE-10 (*Yang et al., 2012*; *Zhang et al., 2012*), and their interactors (*Phillips et al., 2012*; *Uebel et al., 2018*; *Yang et al., 2012*; *Zhang et al., 2012*). These 2° siRNAs are bound by one of several Argonautes (*Yigit et al., 2006*), resulting in the eventual degradation of target mRNAs in the cytoplasm, which requires a cytoplasmic Argonaute, and/or co-transcriptional silencing of the target gene in the nucleus, which requires a nuclear Argonaute (e.g., NRDE-3 *Guang et al., 2008* in somatic cells). Although it is difficult to compare the silencing of two different genes by controlling all relevant variables, past studies have highlighted gene-specific differences in the efficacy of RNAi under different conditions (e.g., when RNAi is enhanced through the loss of the exonuclease ERI-1; *Kennedy et al., 2004*, when nuclear silencing is blocked in somatic cells through

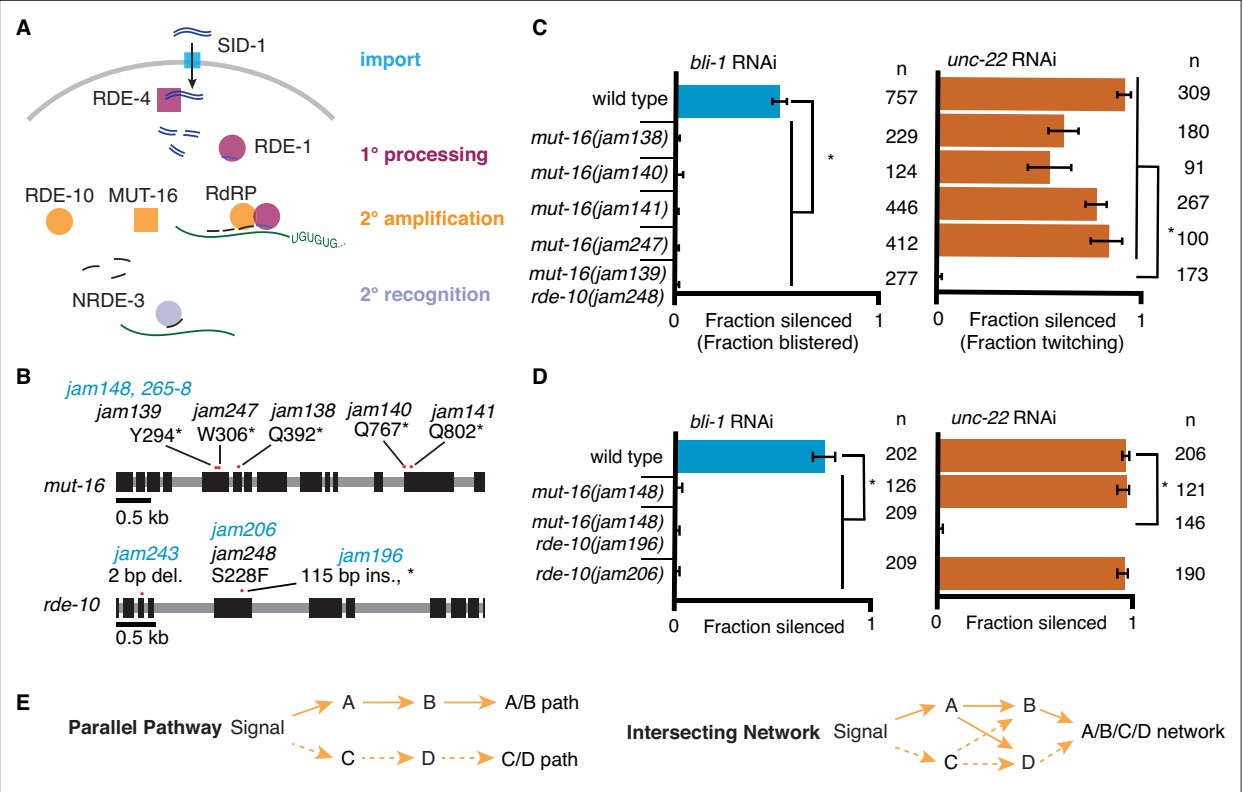

**Figure 1.** RNA interference (RNAi) of two somatic targets shows stark differences in their requirements for MUT-16 and RDE-10. (**A**) Overview of RNAi in somatic cells. Double-stranded RNA (dsRNA, blue) enters the cell through the importer SID-1 (import, teal), after which it is processed by the dsRNA-binding protein RDE-4 and the endonuclease Dicer into 1° short interfering RNAs (siRNAs) that are bound by the primary Argonaute RDE-1 (1° processing, pink). mRNA transcripts (green) recognized by these 1° siRNAs are modified after cleavage by the 3' addition of UG repeats (pUG RNA) and act as templates for the amplification of 2° siRNAs aided by the intrinsically disordered protein MUT-16, the Maelstrom domain-containing protein RDE-10, and RNA-dependent RNA polymerases (2° amplification, orange). These 2° siRNAs can bind secondary Argonaute(s) (e.g., NRDE-3), which can then recognize additional complementary targets (2° recognition) and cause gene silencing. See text for details. (**B**) Gene schematics depicting the mutant alleles found in a genetic screen (black) and/or created using genome editing (blue). Black boxes indicate exons and red dots indicate locations of mutations. Allele names (e.g., *jam139*) and expected amino acid change in the corresponding proteins (e.g., mutation of a tyrosine codon to a stop codon, Y294*) are indicated. See **Figure 1—figure supplement 1** for details of genetic screen. (**C, D**) Response to *bli-1* or *unc-22* RNAi in different mutants. For each mutant, the fraction of animals that showed *bli-1* silencing or *unc-22* silencing (fraction silenced) and the numbers of animals scored (*n*) are shown. Asterisks indicate p < 0.05 for each comparison (brackets) using Wilson's estimates with continuity correction and error bars represent 95% confidence interval. (**C**) Of five isolates with a mutation in *mut-16*, four (*jam138, jam140, jam141,* and *jam247*) failed to silence *bli-1* (blue) but retained *unc-22* silencing (orange). The other mutant failed to silence both genes and additionally had a mutation in *rde-10* (*mut-16(jam139) rde-10(jam248)*). (**D**) Mutants created using genome editing recapitulated the selective silencing of *unc-22* in *mut-16(−)* single mutants (*mut-16(jam148)*) and the failure to silence both genes in *mut-16(−) rde-10(−)* double mutants (*mut-16(jam148) rde-10(jam206)*). Using genome editing to recreate the *jam248* mutation, which is expected to make a mutant protein (RDE-10(S228F)) that disrupts the Maelstrom domain (see **Figure 1—figure supplement 1**), resulted in animals (*rde-10(jam196)*) that showed *unc-22* silencing but not *bli-1* silencing. (**E**) Selective requirement for a regulator could reflect two underlying mechanisms of RNA silencing: (1) Two parallel pathways (left, A/B path vs C/D path) that are differentially used for different target genes; or (2) One intersecting network (right, A/B/C/D network) with quantitative contributions by all regulators along with different thresholds for each target gene.

The online version of this article includes the following source data and figure supplement(s) for figure 1:

**Source data 1.** Excel sheet containing raw data from RNA interference (RNAi) feed depicted in **Figure 1B**.

**Source data 2.** Excel sheet containing raw data from RNA interference (RNAi) feed depicted in **Figure 1D**.

**Figure supplement 1.** A forward-genetic screen identifies a mutation that is expected to disrupt the Maelstrom domain of RDE-10.

**Figure supplement 2.** The predicted structure of the Maelstrom domain of RDE-10 is similar to that of the 3'–5' exonuclease ERI-1.

---

loss of NRDE-3; **Raman et al., 2017**, or when different concentrations of dsRNA are used; **Zhuang and Hunter, 2011**). Understanding the sources of such differences and the underlying mechanisms will improve our ability to design efficacious dsRNA drugs that are difficult to evade through the development of resistance.

Here, we analyze the requirements for silencing two exemplar genes and use quantitative modeling to advance a parsimonious view of RNAi in somatic cells. We show that MUT-16, RDE-10, and NRDE-3 are each required for the silencing of *bli-1*, but any two of these proteins are sufficient for *unc-22* silencing. These differences can be explained by differences in the thresholds for silencing the two genes using an intersecting network of regulators but not by parallel pathways of regulation after primary mRNA recognition. The requirement for NRDE-3 but not for MUT-16 or RDE-10 can be bypassed by enhancing the processing of dsRNA, suggesting that loss of NRDE-3 has the least impact on the efficiency of silencing. A dynamic model of RNA changes during silencing by dsRNA reveals several criteria for efficient RNA silencing. Insights from modeling led us to discover the influence of *cis*-regulatory regions on the requirements for RNAi, the recovery of animals from RNAi within non-dividing cells, and a dearth of pUG RNA production by 2° siRNAs.

## Results
### Two genes with different thresholds for silencing reveal an intersecting network of regulators that mediate RNA interference

To identify regulators of RNA interference (RNAi), we performed a primary screen for mutants that disrupt the maintenance of mating-induced silencing of a transgene followed by a secondary screen for mutants that are also defective in the silencing of endogenous genes by ingested dsRNA (*Figure 1— figure supplement 1A, B*). Mating males with a transgene that expresses fluorescent proteins to hermaphrodites that lack the transgene can initiate silencing in progeny that lasts in descendants for hundreds of generations (*Devanapally et al., 2021*), providing a stable strain that can be mutagenized to look for mutations that result in the recovery of expression from the fluorescent transgene. Of the 15 fertile mutants that showed re-expression, whole-genome sequencing followed by in silico complementation (see Materials and methods), revealed five mutants that had premature stop codons in *mut-16* (*Figure 1B*), a known regulator of RNAi that is required for the production of secondary siRNAs (*Phillips et al., 2012*; *Uebel et al., 2018*; *Zhang et al., 2011*). MUT-16 is detectable in the germline localized within perinuclear foci, but it is also found throughout the soma (*Uebel et al., 2018*). MUT-16 is required for the silencing of all tested somatic targets except the muscle gene *unc-22*, which showed residual silencing ('+++' vs '+' but not '-' in *Zhang et al., 2011*) consistent with its early identification as a sensitive target for RNAi (*Fire et al., 1998*). While all five putative *mut-16* mutants failed to silence the hypodermal gene *bli-1* (*Figure 1C*, *left*), only four of the five showed *unc-22* silencing (*Figure 1C*, *right*). Upon further analysis of the mutant that failed to silence *unc-22*, we found that this mutant also contained a missense mutation in RDE-10, another known regulator of RNAi that is required for the production of secondary siRNAs (*Yang et al., 2012*; *Zhang et al., 2012*). This missense mutation (Ser228Phe) is expected to disrupt the Maelstrom domain of RDE-10 (*Figure 1—figure supplement 1C*), and thus could result in a loss of RDE-10 function. Although the biochemical function of RDE-10 is unknown, it has structural homology with the 3'–5' exonuclease ERI-1 (*Figure 1—figure supplement 2*). To eliminate possible confounding effects of multiple mutations in strains isolated from a genetic screen, we used Cas9-mediated genome editing to introduce mutations in *mut-16* (null) and/or *rde-10* (null or a missense mutation that encodes Ser228Phe) in a wild-type background (*Figure 1B*). While the newly created *mut-16(null)* mutants showed *unc-22* silencing as expected, *mut-16(null) rde-10(null)* (*Figure 1D*, *right*, *Figure 1—figure supplement 2B*) double mutants failed to silence *unc-22*. Since *unc-22* is a particularly sensitive target for RNAi (*Fire et al., 1998*), this lack of *unc-22* silencing in the absence of two regulators with roles in the amplification of 2° siRNAs suggests that 1° siRNA production and RDE-1-mediated recognition of the mRNA is likely not sufficient to cause silencing of most genes. These observations suggest that MUT-16 and RDE-10 are redundantly or additively required for silencing *unc-22* and that the Maelstrom domain of RDE-10 is required for this function. Since the primary Argonaute RDE-1 is required for the silencing of all somatic targets (*Figure 1A*; *Parrish and Fire, 2001*; *Tabara et al., 1999*), including *unc-22*, we propose that MUT-16 and RDE-10 act in parallel downstream of RDE-1 to promote the amplification of 2° siRNA. This branching of the RNAi pathway downstream of RDE-1 could result in strictly parallel pathways where MUT-16 and RDE-10 are used to silence different sets of genes (*Figure 1E*, *left*) or in an intersecting network where both regulators contribute to the silencing of all genes (*Figure 1E*, *right*).

Additional observations suggest differences in the requirements for silencing *bli-1* and *unc-22*. Animals that lack MUT-16 (*Figure 1D*), RDE-10 (*Figure 1D*) or the somatic nuclear Argonaute NRDE-3 (*Raman et al., 2017*) fail to silence *bli-1* but not *unc-22*. However, *rde-10(–); nrde-3(–)* double mutants fail to silence *unc-22* (*Yang et al., 2012*). Therefore, if there were strictly parallel pathways downstream of MUT-16 and RDE-10, then NRDE-3 would be expected to function downstream of MUT-16 but parallel to RDE-10. To test this possibility, we generated *nrde-3(–)* mutants using genome editing (*Figure 2A*) and compared silencing in single mutants and double mutant combinations using the newly generated mutants lacking MUT-16, RDE-10, or NRDE-3. As expected, all single mutants failed to silence *bli-1* but silenced *unc-22*. Surprisingly, all double mutants failed to silence both *bli-1* and *unc-22* (*Figures 1D and 2B*). This requirement for any two of MUT-16, RDE-10, or NRDE-3 suggests that the RNAi pathway cannot be strictly parallel downstream of RDE-1 (see *Figure 2—figure supplement 1A*).

The stark differences in the extents of silencing *bli-1* (~0%) versus *unc-22* (~100%) (*Figures 1D and 2B*) in animals lacking MUT-16, RDE-10, or NRDE-3 suggest that there could be target-specific pathways for silencing, tissue-specific differences in the expressions of RNA regulators, or more parsimoniously, that each regulator contributes to the silencing of both targets as part of an intersecting network, through transcriptional gene silencing and/or post-transcriptional gene silencing (*Figure 2C*), with *unc-22* being more sensitive to silencing than *bli-1*. For such an intersecting network with quantitative contributions by multiple regulators of RNAi to explain the silencing of somatic targets, including targets like *unc-22* and *bli-1* that show dramatic differences, it should be possible to identify values for the relative contributions of each regulatory path (Nm = from MUT-16 to NRDE-3, Nr = from RDE-10 to NRDE-3, Om = from MUT-16 to other Argonautes, and Or = from RDE-10 to other Argonautes in *Figure 2C*, left) and for gene-specific thresholds ($T_{bli-1}$ = level of BLI-1 function below which a defect is detectable, and $T_{unc-22}$ = level of UNC-22 function below which a defect is detectable) that are consistent with all experimental data ('constraints' in *Figure 2C*, right). Of the 100,000 sets of parameters simulated, 145 sets satisfied all experimental constraints (*Figure 2D*). These allowed parameter sets were obtained despite the conservative assumption that the levels of mRNA knockdown for detecting observable defects for *bli-1* and *unc-22* are similar. Relaxing this assumption will lead to a larger number of allowed parameter sets. These valid parameter sets included cases with different relative contributions from RDE-10 and MUT-16 to NRDE-3-dependent silencing for a range of threshold differences for silencing *bli-1* versus *unc-22* (*Figure 2D*, left). Furthermore, extreme contributions of MUT-16 versus RDE-10 (*Figure 2D*, middle) or NRDE-3 versus other Argonautes (*Figure 2D*, right) were absent. Finally, only thresholds for *bli-1* silencing that are less than ~5.5× the threshold for *unc-22* silencing were supported despite the allowed range of up to 100× (*Figure 2D*). Consistent with different quantitative contributions to silencing by each regulator, reducing the availability of *unc-22* dsRNA revealed a graded silencing response: *mut-16(–)<rde-10(–)<nrde-3(–)* (*Figure 2E*; ~5% silencing in *mut-16(jam148)*, ~15% in *rde-10(jam206)*, and ~70% in *nrde-3(jam205)*). Consistent with the possibility of differential contributions from each regulator for different targets (*Figure 2D*), while partial silencing is observable in the absence of NRDE-3 but not MUT-16 or RDE-10 when the muscle gene *unc-54* is targeted (*Figure 2F*, left), partial silencing is observed in the absence of MUT-16 but not NRDE-3 or RDE-10 when the hypodermal gene *dpy-7* is targeted (*Figure 2F*, right).

Taken together, our results are consistent with a single network for RNAi targeting somatic genes where intersecting regulatory pathways downstream of mRNA recognition provide quantitative contributions to silencing.

## The genetic requirement for NRDE-3, but not for MUT-16 and/or RDE-10, can be bypassed by enhancing dsRNA processing

The production of pUG RNAs and 2° siRNAs requires the participation of mRNA (*Figure 1A*), making the contributions of some steps during RNAi gene specific. Therefore, genes could differ in their dependence on proteins required for steps downstream of dsRNA processing and 1° siRNA production. Such differential dependencies could in principle be overcome by increasing the amount of available processed dsRNA and/or 1° siRNA when alternative parallel paths are available (e.g., loss of NRDE-3, MUT-16, or RDE-10 in *Figures 1 and 2*) but not when no alternative paths are available (e.g., loss of both MUT-16 and RDE-10 in *Figure 2B*) or when the increase in dsRNA processing is

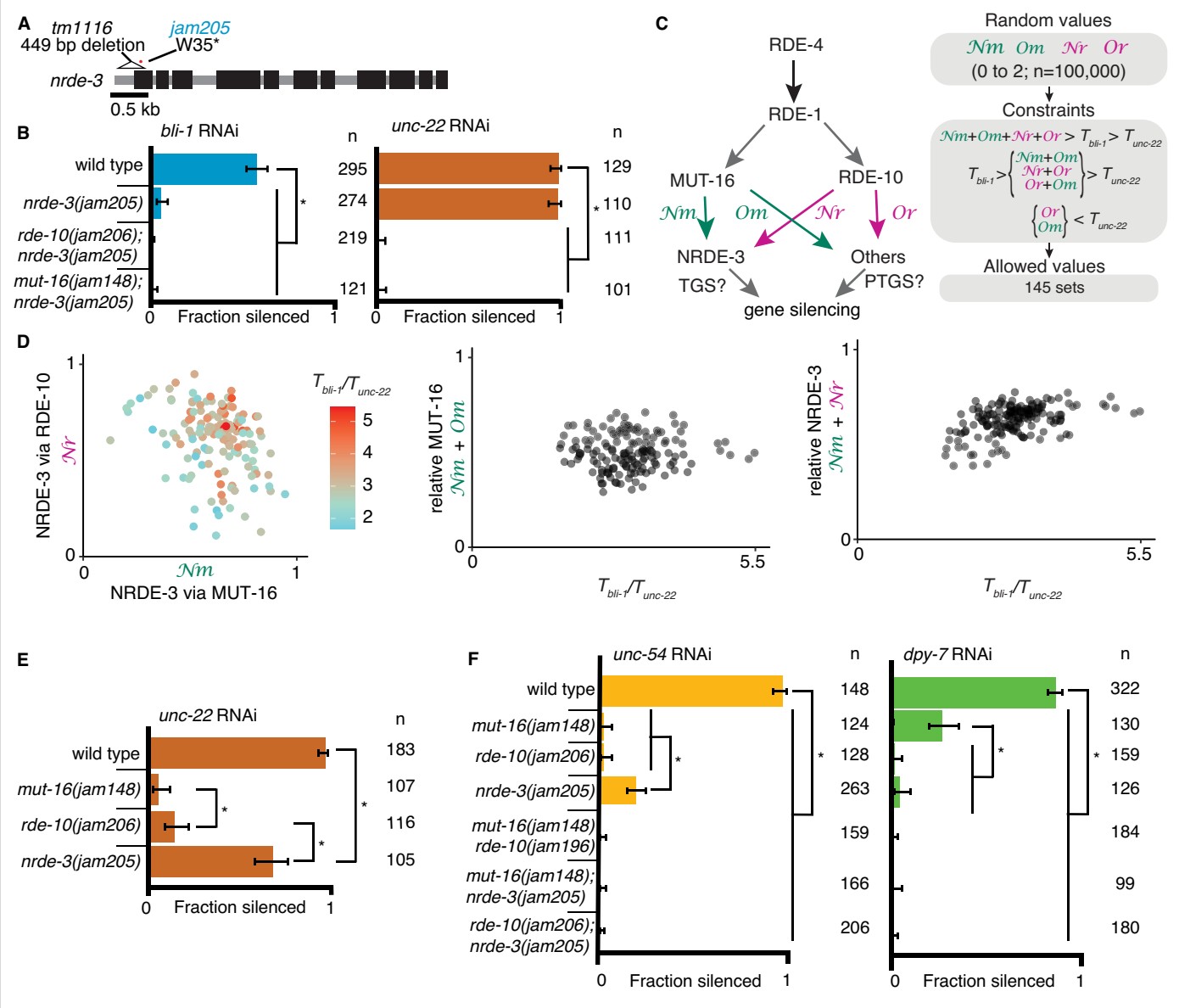

**Figure 2.** Gene-specific requirements and complex redundancy can arise from a single RNA regulatory network. (**A**) Schematic (as in **Figure 1**) depicting *nrde-3* alleles. (**B**) Feeding RNA interference (RNAi) of *bli-1* and *unc-22*. Fractions silenced, numbers scored, comparisons, asterisks, and error bars are as in **Figure 1**. Single mutants lacking NRDE-3 (*nrde-3(jam205)*) fail to silence *bli-1* but not *unc-22*. Double mutants fail to silence both targets. (**C, D**) Mutual constraints among parameters required for a single RNA regulatory network to support experimental results. (**C**, left) Model for a single network of interactors that regulate all RNAi targets in somatic cells. All targets require import (SID-1) and processing (RDE-4 and RDE-1) of double-stranded RNA (dsRNA). Branching after 1° short interfering RNA (siRNA) processing results in four distinct paths (Nm, Or, *Nm*, *Or*) that together contribute to gene silencing, which could occur through co-transcriptional gene silencing (TGS) and/or post-transcriptional gene silencing (PTGS) mechanisms. (**C**, right) Representation of simulation workflow. First, random values between 0 and 2 were drawn for each of the four variables (Nm, Or, *Nm*, *Or*). Second, constraints were added based on the experimental results in (**B**) and **Figure 1D**. Third, allowed values that satisfied all experimental conditions were culled. Of 100,000 sets of random values simulated (0–2 for Nm, Or, *Nm*, *Or* and 0–100 for the ratio of thresholds $T_{bli-1}/T_{unc-22}$), 145 were consistent with all observed responses to RNAi. These allowed numbers reveal the domain of parameter values that support the observed range of gene silencing outcomes using feeding RNAi. (**D**, left) The contribution of NRDE-3 via MUT-16 (Nm) versus that via RDE-10 (Or) for different ratios of thresholds for *bli-1* versus *unc-22* silencing ($T_{bli-1}/T_{unc-22}$) are shown. (**D**, center and right) The relative contributions to silencing that require MUT-16 (Nm + *Nm*, D, *center*) or NRDE-3 (Nm + Or, D, *right*) do not frequently take extreme values and both support a low value for the ratio of thresholds ($T_{bli-1}/T_{unc-22}$ <~5.5 despite allowed values of up to 100). (**E**) Feeding RNAi of *unc-22* assayed as in **Figure 1**, but using aged plates, resulting in weaker RNAi. Animals that lack MUT-16 (*mut-16(jam148)*) have the most severe defect, followed by animals lacking RDE-10 (*rde-10(jam206)*), which is followed by animals lacking NRDE-3 (*nrde-3(jam205)*). (**F**) Feeding RNAi of *unc-54* or *dpy-7*. Fractions silenced, numbers scored, comparisons, asterisks, and error bars are as

*Figure 2 continued on next page*

*Figure 2 continued*

in *Figure 1*. Silencing of *unc-54* showed a partial dependency on NRDE-3, while silencing of *dpy-7* showed a partial dependency on MUT-16, suggesting that the quantitative requirement for a regulator can differ depending on the target.

The online version of this article includes the following source data and figure supplement(s) for figure 2:

**Source data 1.** Excel sheet containing raw data from RNA interference (RNAi) feed depicted in *Figure 2B*.

**Source data 2.** Excel sheet containing raw data from RNA interference (RNAi) feed depicted in *Figure 2E*.

**Source data 3.** Excel sheet containing raw data from RNA interference (RNAi) feed depicted in *Figure 2F*, left (*unc-54*).

**Source data 4.** Excel sheet containing raw data from RNA interference (RNAi) feed depicted in *Figure 2F*, right (*dpy-7*).

**Figure supplement 1.** Support for an intersecting regulatory network.

**Figure supplement 1—source data 1.** Excel sheet containing raw data from RNA interference (RNAi) feed depicted in *Figure 2—figure supplement 1B*.

**Figure supplement 1—source data 2.** Excel sheet containing raw data from RNA interference (RNAi) feed depicted in *Figure 2—figure supplement 1C*.

insufficient. To test these predictions, we increased dsRNA processing and examined silencing in animals lacking different regulators required for the silencing of *bli-1* and/or *unc-22*.

One approach for increasing dsRNA processing is the release of factors such as the endonuclease DCR-1 from competing endogenous pathways by removing the exonuclease ERI-1 (*Lee et al., 2006*). In addition to the increased availability of DCR-1 when ERI-1 is removed, downstream factors involved in siRNA amplification (e.g., MUT-16, MUT-2/RDE-3, RDE-10/11, etc.) and 2° Argonautes (e.g., worm-specific Argonautes (WAGOs)) would be more available to contribute to silencing in response to ingested dsRNA. We used available *eri-1* mutants (*Figure 3A*, *mg366*) and mutants generated using Cas9-mediated genome editing (*Figure 3A*, *jam260* to *jam264*) to test if requirements for silencing *bli-1* and/or *unc-22* could be bypassed. Loss of ERI-1 enabled *bli-1* silencing in animals lacking NRDE-3, but not in animals lacking RDE-10 or MUT-16 (*Figure 3B*). Furthermore, loss of *eri-1* was not sufficient for the complete rescue of *unc-22* silencing in animals lacking any two of these three regulators (*Figure 3C*). An alternative approach for increasing dsRNA processing is the overexpression of the dsRNA-binding protein RDE-4, which recruits dsRNA for processing by DCR-1 (*Tabara et al., 2002*, *Parker et al., 2006*). Minimal amounts of RDE-4 can support RNAi as evidenced by silencing in *rde-4(−)* adult progeny of *rde-4(+/−)* hermaphrodites (Figure S7E in *Marré et al., 2016*) and in *rde-4(−)* animals with trace levels of ectopic expression from multicopy *rde-4(+)* transgenes (Figure 2 in *Raman et al., 2017*). We found that even hemizygous males expressing *rde-4(+)* from a single-copy transgene driving expression in the germline and the intestine under the control of the *mex-5* promoter (*Marré et al., 2016*) was sufficient for rescuing both *bli-1* and *unc-22* silencing (*Figure 3D*). Similar expression of *rde-1(+)*, however, was not sufficient for rescuing silencing in *rde-1(−)* animals (*Figure 3D*), suggesting that small amounts of RDE-4 but not RDE-1 are sufficient for RNAi. RDE-4 can be selectively overexpressed in the hypodermis using a single-copy transgene with a *nas-9* promoter (overexpression evident in *Figure 3—figure supplement 1*; and selectivity demonstrated in Figure 4C in *Raman et al., 2017*). This hypodermal expression of *rde-4(+)* was sufficient to enable *bli-1* silencing in an otherwise *rde-4(−); nrde-3(−)* background (*Figure 3E*). Thus, either loss of ERI-1 or overexpression of RDE-4 can bypass the need for NRDE-3 for silencing *bli-1*, suggesting that the requirement for this regulator does not reflect a specific need for a particular regulator (NRDE-3) but reflects a larger amount of silencing signals required for reducing *bli-1* function sufficiently to cause a detectable defect. However, loss of ERI-1 and/or overexpression of RDE-4 could not compensate for the loss of RDE-10 or MUT-16 for *bli-1* silencing (*Figure 3F*), suggesting that these regulators make a more substantial contribution to *bli-1* silencing than NRDE-3. These observations further support the idea that 1° siRNAs alone are not sufficient to cause silencing, consistent with the lack of *unc-22* silencing in *mut-16(−) rde-10(−)* double mutants (*Figure 1B, D*). One explanation for these results is that in *eri-1(−); nrde-3(−)* double mutants (*Figure 3G*), a different 2° Argonaute is able to compensate for the lack of NRDE-3, whereas in *mut-16(−); eri-1(−)* or *rde-10(−); eri-1(−)* double-mutants, the reduction of 2° siRNAs is too great to cause a detectable Bli-1 defect.

Taken together, these results suggest that gene-specific requirements for some proteins that function in RNAi do not reflect different pathways for silencing different genes, but rather a quantitative requirement for regulators acting as part of an intersecting RNA regulatory network.

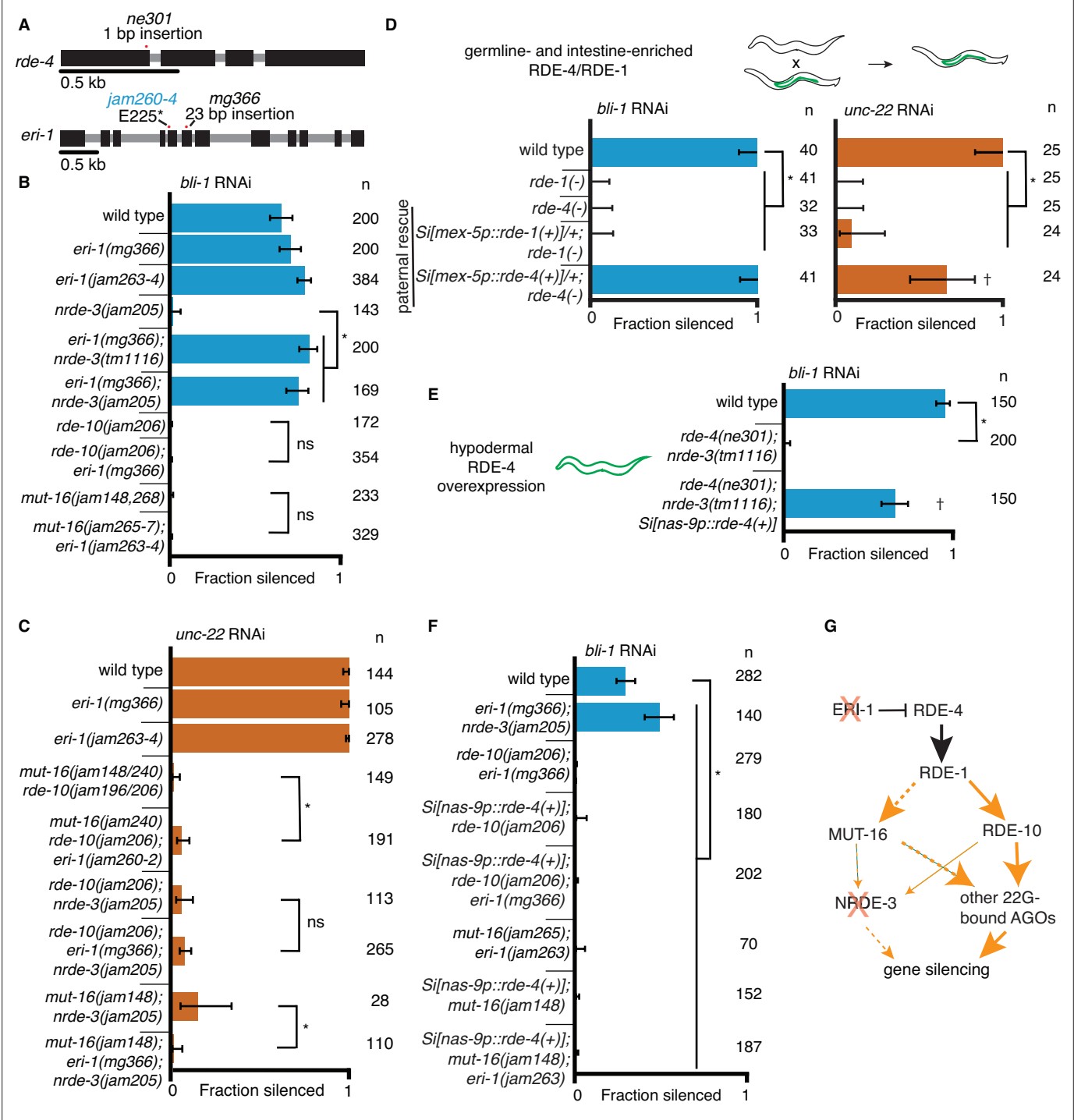

**Figure 3.** Gene-specific requirements for NRDE-3 can be bypassed in two ways. (**A**) Gene schematics (as in *Figure 1*) of *rde-4* and *eri-1*. (**B, C**) Loss of ERI-1 can bypass the NRDE-3 requirements for silencing *bli-1* but not the other requirements for silencing *bli-1* or *unc-22*. Feeding RNA interference (RNAi) targeting *bli-1* (**B**) or *unc-22* (**C**) with fractions silenced, numbers scored, comparisons, asterisks, and error bars as in *Figure 1*. (**B**) Loss of ERI-1 (*mg366*, *jam263*, and *jam264* alleles) can compensate for the role of NRDE-3 (*tm1116* and *jam205* alleles) but not of RDE-10 (*jam206* allele) or MUT-16 (*jam148*, *jam265*, *jam266*, *jam267*, and *jam268* alleles) in *bli-1* silencing. See *Table 3* for additional information. (**C**) Silencing of *unc-22* is not restored by loss of ERI-1 (*mg366*, *jam260*, *jam261*, and *jam262* alleles) in mutants that also lack any two of *mut-16* (*jam148* and *jam240* alleles), *rde-10* (*jam196* and *jam206* alleles), or *nrde-3* (*jam205* allele). See *Table 3* for additional information. (**D, E**) Overexpression of RDE-4 in the hypodermis can bypass the requirement for NRDE-3 in *bli-1* silencing. (**D**) Minimal amounts of RDE-4 are sufficient for somatic silencing. (*Top*) Schematic depicting generation of male progeny with paternal inheritance of a single-copy transgene (*Si[...]*) that expresses *rde-4(+)* or *rde-1(+)* under the control of the *mex-5* promoter

*Figure 3 continued on next page*

*Figure 3 continued*

(*mex-5p*) in the germline (green) of *rde-4(−)* or *rde-1(−)* animals, respectively (germline- and intestine-enriched RDE, based on rescue of RNAi in *rde-1(−)* animals (39)). (*Bottom*) Male cross progeny with the transgene were scored after feeding only F1 animals, showing that unlike animals with germline- and intestine-enriched RDE-1, animals with similarly enriched RDE-4 can rescue both *unc-22* and *bli-1* silencing. Thus, small amounts of RDE-4 potentially mis-expressed in the hypodermis or a non-autonomous effect of RDE-4 from the germline or intestine is sufficient for silencing in the muscle and hypodermis. † indicates $p < 0.05$ when compared to either wild type or the *rde-4(−)* mutant and other symbols are as in (**B**). (**E**) Silencing of *bli-1* is restored in *nrde-3(tm1116); rde-4(ne301)* double mutants when *rde-4(+)* is overexpressed in the hypodermis (*Si[nas-9p::rde-4(+)]*). (**F**) Silencing of *bli-1* is not restored in animals lacking MUT-16 or RDE-10, despite the overexpression of RDE-4 in the hypodermis and/or the loss of ERI-1. (**G**) Summary depicting the bypass of NRDE-3 when ERI-1 is eliminated and/or RDE-4 is overexpressed. The increase in double-stranded RNA (dsRNA) processing increases the contributions of NRDE-3-independent paths to silencing.

The online version of this article includes the following source data and figure supplement(s) for figure 3:

**Source data 1.** Excel sheet containing raw data from RNA interference (RNAi) feed depicted in *Figure 3B*.

**Source data 2.** Excel sheet containing raw data from RNA interference (RNAi) feed depicted in *Figure 3C*.

**Source data 3.** Excel sheet containing raw data from RNA interference (RNAi) feed depicted in *Figure 3D*.

**Source data 4.** Excel sheet containing raw data from RNA interference (RNAi) feed depicted in *Figure 3E*.

**Source data 5.** Excel sheet containing raw data from RNA interference (RNAi) feed depicted in *Figure 3F*.

**Figure supplement 1.** Overexpressing RDE-4 in the hypodermis.

**Figure supplement 1—source data 1.** Unlabeled gel image of semi-quantitative polymerase chain reaction (PCR) for *rde-4* with the addition of reverse transcriptase – depicted in the first row of *Figure 3—figure supplement 1*.

**Figure supplement 1—source data 2.** Unlabeled gel image of semi-quantitative polymerase chain reaction (PCR) for *rde-4* without the addition of reverse transcriptase – depicted in the second row of *Figure 3—figure supplement 1*.

**Figure supplement 1—source data 3.** Unlabeled gel image of semi-quantitative polymerase chain reaction (PCR) for *tbb-2* with the addition of reverse transcriptase – depicted in the third row of *Figure 3—figure supplement 1*.

**Figure supplement 1—source data 4.** Unlabeled gel image of semi-quantitative polymerase chain reaction (PCR) for *tbb-2* without the addition of reverse transcriptase – depicted in the fourth row of *Figure 3—figure supplement 1*.

**Figure supplement 1—source data 5.** Labeled images of all gels used in *Figure 3—figure supplement 1*.

**Figure supplement 1—source data 6.** Excel sheet with quantifications of bands depicted in *Figure 3—figure supplement 1* to obtain the numbers depicted below the cropped gel.

## Quantitative modeling of RNA interference and mRNA production provides rationales for a variety of target-specific outcomes

The many protein regulators of RNAi drive changes in RNA metabolism, including the production of new RNA species (1° siRNA, 2° siRNA, and pUG RNA), that are associated with the targeted gene. Although these changes can be indicators of RNA silencing, the quantitative relationship between such RNA intermediates and the extent of gene silencing measured as a reduction in function of the targeted gene or its mRNA levels is unclear. A priori, reduction in the mRNA levels of a gene could depend on universal processing of imported dsRNA, production of secondary small RNAs with the participation of gene-specific mRNAs, and downregulation of pre-mRNA and/or mRNA influenced by pre-existing gene-specific RNA metabolism. To understand how these gene-specific factors could influence RNA silencing, we began by analyzing the impact of a few characteristics of a gene on mRNA ($m$) and pre-mRNA levels ($p$) after RNAi. We first used a sequential equilibrium model, where we assume each step must be completed before beginning the next, for example dsRNAs are fully processed into 1° siRNAs, then 1° siRNAs can recognize transcripts to result in the production of pUG RNAs, and so on (*Figure 4—figure supplement 1A* and Supplemental Methods). We tested parameters that would result in varying levels of target RNA knockdown (790 of 1 million simulated parameters resulted in $[m]_i < [m]$, $[m]_i > 0$, and $[p]_i > 0$; *Figure 4—figure supplement 1B*). Under this simple model, we found that (1) RNAi can result in different residual concentrations of RNAs for different genes (*Figure 4—figure supplement 1C*); (2) for a given gene, silencing can alter the ratio of pre-mRNA to mRNA (*Figure 4—figure supplement 1D, E*); and (3) effective targeting of mRNA by primary or secondary small RNAs is required for strong silencing (*Figure 4—figure supplement 1F*). These observations hint at the influence of gene-specific factors on the functional outcome of RNAi and impel the exploration of a more detailed dynamic model.

A qualitative outline of the molecular mechanism for RNAi in *C. elegans* has been deduced based on a variety of studies over the last two decades (mechanism outline; *Figure 4A*, *left*), but the cellular, subcellular, and kinetic details of every step remain obscure. Quantitative modeling of RNAi – or indeed any process of interest – could be done at many scales (*Figure 4A*, *right*) based on the level of understanding sought and experimental data available for testing predictions. For example, the responses of different animals in a population to dsRNA exposure (population model; *Figure 4A*, *right*) or the changes in key RNA species after entry of dsRNA into the cytosol (process model; *Figure 4A*, *right*) could be modeled. At yet greater detail, one step such as the amplification of small RNAs using pUG RNA templates could be modeled by incorporating sequence bias, processivity of RdRP, etc. (biochemical model; *Figure 4A*, *right*). Of these scales, we focused on the process model because early process models of RNAi (e.g., *Bergstrom et al., 2003*) were proposed before crucial discoveries on the biogenesis of 2° siRNAs without forming long dsRNA (*Pak and Fire, 2007*) and the stabilization of mRNA templates as pUG RNAs (*Preston et al., 2019*; *Shukla et al., 2020*). Therefore, we incorporated these recent developments and modeled how the addition of dsRNA could disrupt the steady-state RNA metabolism of the targeted gene using ordinary differential equations (*Figure 4B*). While there are many parameters that one could include in any model, we have used a conservative set of parameters for looking at the overall RNAi process without explicitly modeling sub-steps in detail. For example, production of 22G RNA is modeled as a single step rather than one that incorporates how the frequency of Cs in template mRNA, the subcellular localization of mRNA, secondary structure formation in mRNAs, etc. impact the efficiency of silencing. Similarly, genome sequence and its effect on transcription and/or splicing are modeled as a single step, rather than one that looks at frequency of repeats, sizes of introns, chromatin environment, etc. We expect that future studies will build upon this initial model hand-in-hand with the more sophisticated experiments needed to test such detailed hypotheses.

In this initial model (*Figure 4B*), the steady-state levels of pre-mRNA and mRNA – which depend on production, maturation, and turnover – could be altered upon the addition of matching dsRNA through the generation of new RNA species (1° siRNA, 2° siRNA, pUG RNA) that are also subject to turnover. To accommodate these known intermediates and interactions, we used six differential equations to describe the rate of change of key RNA species (dsRNA ($ds$), 1° siRNA ($pri$), pUG RNA ($ug$), 2° siRNA ($sec$), pre-mRNA ($p$), and mRNA ($m$)) with rate or binding constants for different processes ($k1$–$k9$), turnover rates for different RNAs ($T_{pri}$, $T_{ug}$, $T_{sec}$, $T_p$, $T_m$), and variables for the lengths of RNAs ($l_{ds}$ – dsRNA; $l_m$ – mRNA). For example, the rate of change over time for 1° siRNA is modeled as

$$k1. \left[ds\right] . \frac{l_{ds}}{22} - k2. \left[pri\right] . \left[m\right] - T_{pri}. \left[pri\right]$$

which includes the idea that 1° siRNAs are cleaved into 22nt sequences at a certain rate dependent on the dsRNA concentration ($k1. \left[ds\right] . \frac{l_{ds}}{22}$), the amount of 1° siRNAs that can bind target transcripts ($k2. \left[pri\right] . \left[m\right]$), and the turnover of the 1° siRNAs ($T_{pri}. \left[pri\right]$), which notably is dependent on the concentration of the 1° siRNAs.

To illustrate the relative dynamics of different RNA species upon the addition of dsRNA, we computed the concentrations of dsRNA, 1° siRNA, pUG RNA, 2° siRNA, pre-mRNA, and mRNA using the equations after assigning arbitrary values for the different constants (*Figure 4C*; see legend for parameter values). To account for the non-negative values of all RNA species within cells, we ensured that the values of incremental change $dx$ for any species $x$ was only added if ($x + dx$) >0 and set to be 0 if ($x + dx$) ≤ 0. This bounding of the rate equations allows for any approach to zero. As expected, the levels of dsRNA decrease (*Figure 4C*, red) as it is processed into 1° siRNA (*Figure 4C*, purple), which eventually decays because of turnover. This transient accumulation of 1° siRNA is followed by that of pUG RNAs (*Figure 4C*, green) and of 2° siRNA (*Figure 4C*, brown). Silencing of the target is reflected in the lowered levels of mRNA (*Figure 4C*, blue) and pre-mRNA (*Figure 4C*, orange). However, these levels eventually recover upon turnover of the silencing intermediates (1° siRNA, pUG RNA, 2° siRNA). Although we assumed the turnover of 1° siRNA, 2° siRNA, and pUG RNA for the modeling, the experimental demonstration of recovery (either of individual RNA species or of the entire phenotype) from RNA silencing in non-dividing cells would be needed to support the existence of such turnover mechanisms for these different types of RNAs.

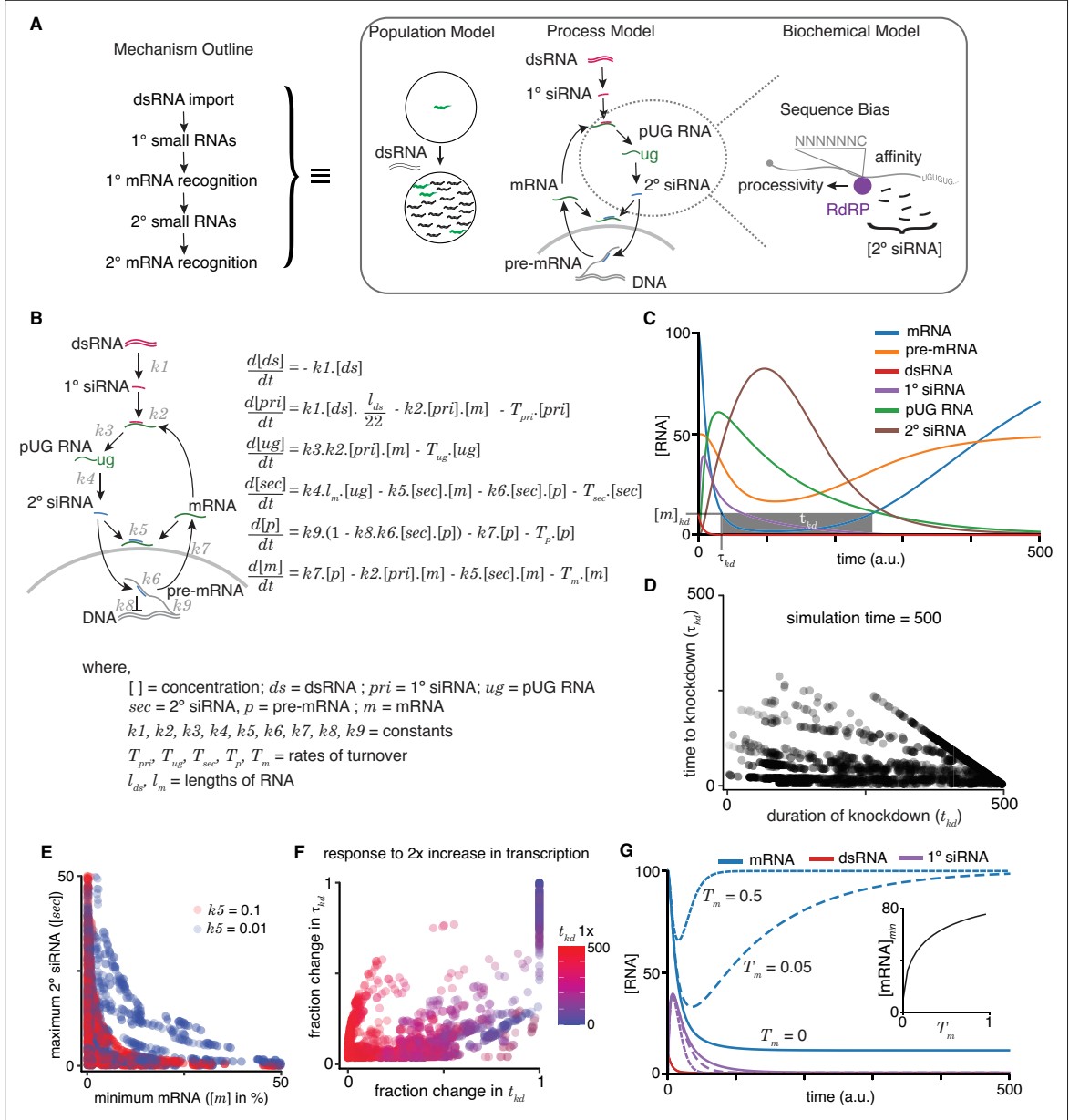

**Figure 4.** A quantitative model allows exploration of parameters for RNA interference (RNAi) in *C. elegans*. (**A**) An outline of the mechanism of RNAi (*left*) has been deduced from experiments dissecting the response of animal populations, changes in RNA populations, and biochemical sub-steps (e.g., 2° short interfering RNA [siRNA] production using pUG RNA), each of which can be separately modeled (*right*). The modeling in this study focuses on the overall RNA-mediated processes that accompany RNAi. (**B**) Schematic (*left*) and ordinary differential equations (*right*) describing the production and turnover of different RNA species. Rate constant for 1° siRNA processing from double-stranded RNA (dsRNA) (*k1*), binding constant for 1° siRNA-binding target mRNA (*k2*), rate constant for pUG RNA production (*k3*), rate constant for 2° siRNA production (*k4*), binding constant for 2° siRNAs binding mRNA (*k5*) or pre-mRNA (*k6*), rate constant for export and splicing of transcripts from the nucleus (*k7*), rate constant for repression of transcription (*k8*), and rate constant for new transcript production (*k9*). Other terms are described in the figure. (**C**) Relative changes in the concentrations of each RNA ([RNA] vs time in a.u.; dsRNA, mRNA, pre-mRNA, pUG RNA, 1° siRNA, and 2° siRNA) for an example set of parameters (all turnover rates = 0.05, $k1 = 1$, $k2 = 0.01$, $k3 = 1$, $k4 = 0.05*l_m = 0.5$, $k5 = 0.01$, $k6 = 0.01$, $k7 = 0.1$, $k8 = 0.05$, $k9 = 7.5$) are illustrated. A reduction to 10% of initial mRNA concentration is designated as the threshold for detecting a defect upon knockdown ($[m]_{kd}$), the time needed to reach the threshold ($\tau_{kd}$) and the time for which mRNA levels remain below the threshold ($t_{kd}$) are also indicated. (**D**) Relationship between the duration of knockdown and the time to knockdown ($t_{kd}$ and $\tau_{kd}$ are as in (**C**)). (**E**) Relationship between mRNA concentration and 2° siRNA accumulation. The minimum mRNA concentrations and maximum 2° siRNA concentrations reached for different transcripts with two different binding constants of 2° siRNAs binding to mRNA ($k5 = 0.1$, red and $k5 = 0.01$, blue) are plotted. Also see *Figure 4—figure supplement 2*. (**F**) Impact of doubling transcription on transcripts with different knockdown parameters. Each transcript is colored based on its initial duration of knockdown ($t_{kd}$, blue to red gradient) before a twofold increase in the rate constant for transcription (*k9*) and the resultant fractional change in the duration of knockdown $t_{kd}$ is plotted against that in the time to knockdown $\tau_{kd}$. (**G**) Genes

*Figure 4 continued on next page*

*Figure 4 continued*

with higher turnover are harder to knockdown. Response of mRNAs and their respective 1° siRNA with the same steady-state concentrations but with different rates of mRNA turnover (solid lines: $T_m = 0$, large dashes: $T_m = 0.05$, small dashes: $T_m = 0.5$) upon addition of 10 molecules of dsRNA are shown. (*Inset*) Relationship of the minimum concentration of mRNA ([mRNA]$_{min}$) with its $T_m$ in response to a fixed amount of dsRNA.

The online version of this article includes the following figure supplement(s) for figure 4:

**Figure supplement 1.** An equilibrium model for RNA interference (RNAi).

**Figure supplement 2.** Impact of model parameters on the time to knockdown ($\tau_{kd}$) and the duration of knockdown ($t_{kd}$).

For any gene, the time to knockdown ($\tau_{kd}$) and the duration of knockdown ($t_{kd}$) could be used to evaluate the efficiency of RNAi (knockdown = 10% of initial mRNA concentration in *Figure 4C*). The different RNA species made downstream of 1° RNA binding in *C. elegans* provide the opportunity for multiple parameters to differ between genes. Therefore, we varied each parameter and examined $\tau_{kd}$ and $t_{kd}$ as indicators of efficiency (*Figure 4—figure supplement 2*). Overall, $\tau_{kd}$ and $t_{kd}$ were uncorrelated (*Figure 4D*), with cases of rapid but transient knockdown, which would necessitate multiple dosing of dsRNA for sustained effects. While loss of function through the reduction of mRNA levels is often the intended goal of knockdown, RNA intermediates could serve as convenient and quantitative measures of molecular changes. For example, the abundant 2° siRNAs have been a widely used molecular indicator of silencing (e.g., *Gu et al., 2009*). However, the maximal amount of 2° siRNAs that accumulate is not correlated with strong silencing as measured by the minimal amount of mRNA during knockdown (*Figure 4E*). Additionally, an increase in transcription generally resulted in poorer knockdown through changes in both $\tau_{kd}$ and $t_{kd}$ (*Figure 4F*), consistent with the obvious expectation that a gene with transcriptional upregulation during exposure to dsRNA will be more difficult to knockdown.

Efficient silencing using dsRNA is possible in many organisms, including mammals, despite silencing relying on mostly post-transcriptional degradation of mRNA without the production of pUG RNA or 2° siRNA (*Sandy et al., 2005*). To explore differences between genes that could impact the efficiency of RNA silencing universally in any system, we simulated knockdown through the post-transcriptional loss of mRNA alone by eliminating production of pUG RNAs, and thus downstream secondary small RNAs and transcriptional silencing (*Figure 4B*, $k3 = 0$). When a fixed amount of dsRNA was exposed to different genes with the same amount of mRNA at steady state, genes with higher mRNA turnover rates showed less efficient knockdown (*Figure 4G*). This inverse relationship is expected because to maintain the same steady-state levels, genes with higher mRNA turnover must also have higher mRNA production. As a result, for the same amount of added dsRNA and the same steady-state level of mRNA before exposure to dsRNA, the mRNA levels will recover faster for genes with higher production coupled with higher turnover.

In summary, varying a few gene-specific parameters clarified the diversity of outcomes that are possible in response to the same dose of dsRNA. Gene-specific differences make the time to knockdown and the duration of knockdown uncorrelated and reduce the utility of key intermediates of RNA silencing as predictors of knockdown efficiency. Increases in transcription during exposure to dsRNA and high turnover of mRNA coupled with high production at steady state reduce the efficiency of knockdown. While the predictions of the model include quantitative relationships that will require advances in the quantitative measurement of many steps during RNAi, the model also makes some qualitative predictions that can be immediately tested.

## Changing *cis*-regulatory elements of a gene impacts its requirements for silencing by dsRNA

A key realization from the exploration of the dynamic model for RNAi is that pre-existing RNA regulation of a gene impacts the response to dsRNA (*Figure 4*). However, the individual impacts of the many features of a gene that together set its RNA metabolism (e.g., promoter, 3' *cis*-regulatory regions, introns, genomic location, etc.) are usually unknown. Nevertheless, as tests of this possibility, we altered target genes using Cas9-mediated genome editing and examined changes, if any, in the genetic requirement for NRDE-3 for silencing by ingested dsRNA.

First, we swapped the 3' *cis*-regulatory regions of *bli-1* and *unc-22* (*Figure 5A*, *Figure 5—figure supplement 1A*). Animals with the *unc-22 3'cis* sequence in place of the *bli-1 3'cis* (*bli-1p::bli-1::unc-22*

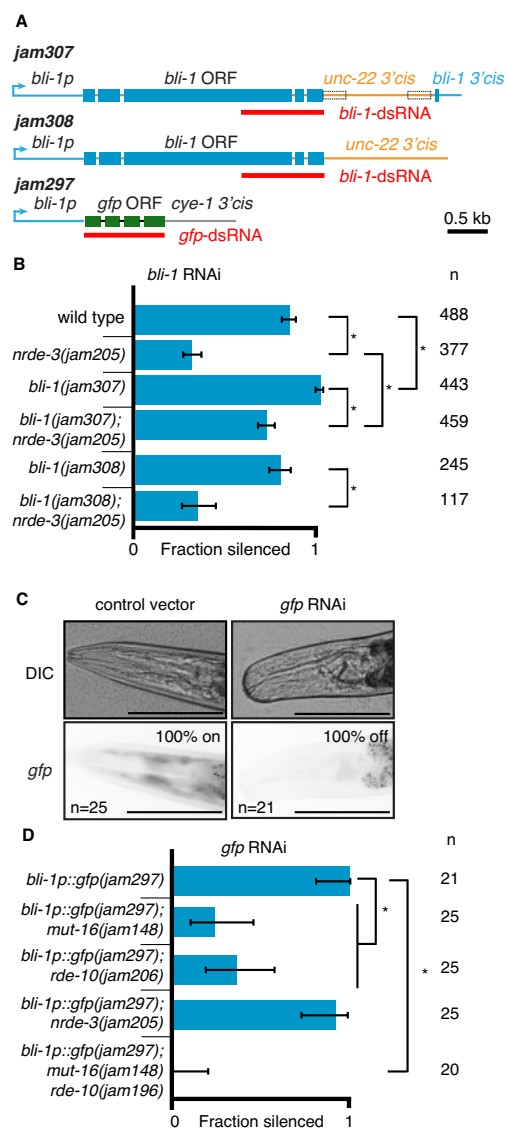

**Figure 5.** Changing *cis*-regulatory regions or open reading frames can alter the observed dependence on NRDE-3 for silencing by double-stranded RNA (dsRNA). (**A**) Schematics (as in *Figure 1*) of the hybrid genes created to test the role of 3' and 5' *cis*-regulatory sequences in the response to dsRNA (see Methods for details). (**B**) The need for NRDE-3 to cause silencing using *bli-1*-dsRNA is reduced when the 3' *cis*-regulatory sequences of *bli-1* are replaced with that of *unc-22* (*bli-1(jam307)*). Fractions silenced, numbers scored, comparisons, asterisks, and error bars are as in *Figure 1* (**C**) Representative images (DIC, top; GFP fluorescence, bottom) of the head region of *bli-1p::gfp::cye-1 3'cis* animals either in response to a control vector (pL4440, *left*) or in response to *gfp*-dsRNA (*right*). The presence or absence of *gfp* in the head region was used to score the fraction silenced in (**D**). Scale bar = 100 μm. (**D**) Use of the *bli-1* promoter alone does not confer the genetic requirements for

*Figure 5 continued on next page*

*Figure 5 continued*

silencing *bli-1*. Silencing of GFP expressed from the *bli-1p::gfp::cye-1 3'cis* transgene (*bli-1p::gfp(jam297)*) by *gfp*-dsRNA was measured in a wild-type background, in animals lacking NRDE-3 (*nrde-3(jam205)*), in animals lacking RDE-10 (*rde-10(jam206)*), in animals lacking MUT-16 (*mut-16(jam148)*), and in animals lacking both MUT-16 and RDE-10 (*mut-16(jam148) rde-10(jam196)*). Fractions silenced, numbers scored, comparisons, asterisks, and error bars are as in *Figure 1*.

The online version of this article includes the following source data and figure supplement(s) for figure 5:

**Source data 1.** Excel sheet containing raw data from RNA interference (RNAi) feed depicted in *Figure 5B*.

**Source data 2.** Representative images of *bli-1p::gfp* transgenic worms in response to control or *gfp* RNA interference (RNAi) in *Figure 5C* (subset from images in *Figure 5—source data 4*).

**Source data 3.** Excel sheet containing raw data from RNA interference (RNAi) feed depicted in *Figure 5D*.

**Source data 4.** All images used to score the RNA interference (RNAi) feed depicted in *Figure 5D*.

**Figure supplement 1.** Replacing the *unc-22* 3'*cis*-regulatory sequence with that of *bli-1* results in defects that show a NRDE-3-dependent enhancement when exposed to *unc-22*-dsRNA.

**Figure supplement 1—source data 1.** Excel sheet containing raw data from RNA interference (RNAi) feed depicted in *Figure 5—figure supplement 1B*.

---

*3'cis*) showed a much reduced dependence on NRDE-3 (*Figure 5B*), Animals with the *bli-1 3'cis* sequence downstream of the coding sequence of *unc-22* (*unc-22p::unc-22::bli-1 3'cis*) showed substantial twitching even without *unc-22* RNAi (*Figure 5—figure supplement 1B*). Yet, we were able to discern an enhancement upon addition of *unc-22* dsRNA. This enhancement was absent in animals lacking NRDE-3. Together, these results provide evidence for prior regulation (potentially mediated via the 3' UTR) impacting the genetic requirements for silencing.

Next, we used the *bli-1* 5' *cis*-regulatory regions (promoter) and the cyclin E 3' *cis*-regulatory regions to drive *gfp* expression in hypodermal cells (*Figure 5A*; *bli-1p::gfp::cye-1 3'cis*). Animals with this gene showed expression of GFP in the hypodermis, which is most easily visible in the head region and is detectably silenced by ingested *gfp*-dsRNA in a wild-type background (*Figure 5C*). Similar silencing was detectable in animals lacking NRDE-3 (*Figure 5D*), but the silencing in animals lacking MUT-16 or RDE-10 was much weaker (*Figure 5D*). These observations are in contrast to the observed lack of detectable

silencing of the wild-type *bli-1* gene in response to *bli-1*-dsRNA in animals lacking NRDE-3, MUT-16, or RDE-10 (*Figures 1D and 2B*), suggesting that the *bli-1* promoter is not sufficient to confer these requirements on all genes. Together, these results reveal that two different genes expressed under the same promoter within the same tissue can have different requirements for silencing in response to dsRNA (*bli-1* vs *gfp* under the control of *bli-1p*).

These initial attempts to change the requirements for the response to dsRNA by altering the pre-existing regulation of target genes encourage the exploration of additional factors predicted to differentially influence RNA silencing of different genes (*Figure 4*, *Figure 4—figure supplement 2*).

## Gene expression can recover after knockdown despite the presence of amplification mechanisms

The dynamic model (*Figure 4*) assumes that all key RNA intermediates (1° siRNA, 2° siRNA, and pUG RNA) are subject to turnover. If this assumption is true, animals should be able to recover from RNA silencing in non-dividing cells despite the production of abundant 2° siRNAs using RNA-dependent RNA polymerases. Experimental detection of the re-establishment of wild-type phenotype after a pulse of RNAi would provide evidence not only for the recovery of mRNA levels but also the subsequent production of functional protein. To test this possibility, we exposed wild-type animals to a 1-hr pulse of dsRNA matching the sensitive target *unc-22* and examined them for the Unc-22 defect every 24 hr (*Figure 6A*). With this limited exposure to dsRNA, we observed only ~80% silencing after the first 24 hr, which reached ~100% by day 3, suggesting that it takes a couple of days after exposure to small amounts of dsRNA to observe complete silencing. This delay could be driven by the time required for the buildup of RNA intermediates required for silencing (1° siRNA, 2° siRNA, and/or pUG RNA), for the turnover of UNC-22 protein, and/or for the dissipation of events downstream of the molecular role of UNC-22. Consistent with recovery, silencing was only observed in ~50% of the animals on day 5, which dropped to ~36% by the eighth day after RNAi. In contrast, animals that were continually fed *unc-22* RNAi showed ~100% silencing even at day 7 (*Figure 6A*), suggesting that the RNAi machinery remains functional in aging animals. Since the body-wall muscle cells – where *unc-22* functions – do not divide during adulthood (*Krause and Liu, 2012*), this observation of recovery from silencing cannot be explained by the dilution of dsRNA and/or RNA intermediates through cell division. Thus, these results support the turnover of all key RNA intermediates generated during RNAi – 1° siRNA, 2° siRNA, and pUG RNA, and highlights for the first time that a target gene can recover from RNAi even in non-dividing cells despite their use of RNA-dependent RNA polymerases to amplify silencing signals.

## Sequences in mRNA that match the trigger dsRNA are hot zones of pUG RNA production

Of the RNA intermediates generated during RNAi, pUG RNAs have been proposed to be used as stable templates to produce small RNAs (*Shukla et al., 2020*). Sustained production of small RNAs could occur if the targeting of mRNA by 2° siRNA resulted in further pUG RNA production, subsequent 3° siRNA production, and so on, thereby providing a way for silencing to persist despite the turnover of all RNA species. However, the production of such 3° siRNA has been observed only when targeting a germline gene (*Sapetschnig et al., 2015*) and not when targeting a somatic gene (*Pak et al., 2012*). To examine whether such repeated rounds of pUG RNA production occur during RNAi of *unc-22*, we fed wild-type worms bacteria that express *unc-22* dsRNA or control dsRNA (L4440) and looked for the presence of pUG RNAs. These RNAs are detected as a heterogenous mixture using RT-PCR with a poly-CA 3′ primer and gene-specific 5′ primers. Consistent with the production of pUG RNAs upon targeting by 1° siRNAs, we detected pUG RNAs generated after cleavage within the *unc-22* mRNA sequence that matches the dsRNA (*Figure 6B*, 0 kb 5′ primer). Since 2° siRNAs are made with a 5′ bias on the mRNA template (*Pak et al., 2012*; *Pak and Fire, 2007*), pUG RNAs generated in response to targeting by 2° siRNAs are expected to include mRNAs cleaved upstream of the sequence matching the dsRNA. Surprisingly, all pUG RNAs detected using a 5′ primer ~1 kb upstream of the target sequence were larger than 1 kb (*Figure 6C*, 1 kb 5′ primer), suggesting that there is a dearth of pUG RNA formation through cleavage within 1 kb upstream of sequences targeted by dsRNA. Notably, this absence is despite the expected relative ease of amplifying shorter sequences when compared with amplifying longer sequences using the same primers. This lack of detectable pUG RNAs upstream

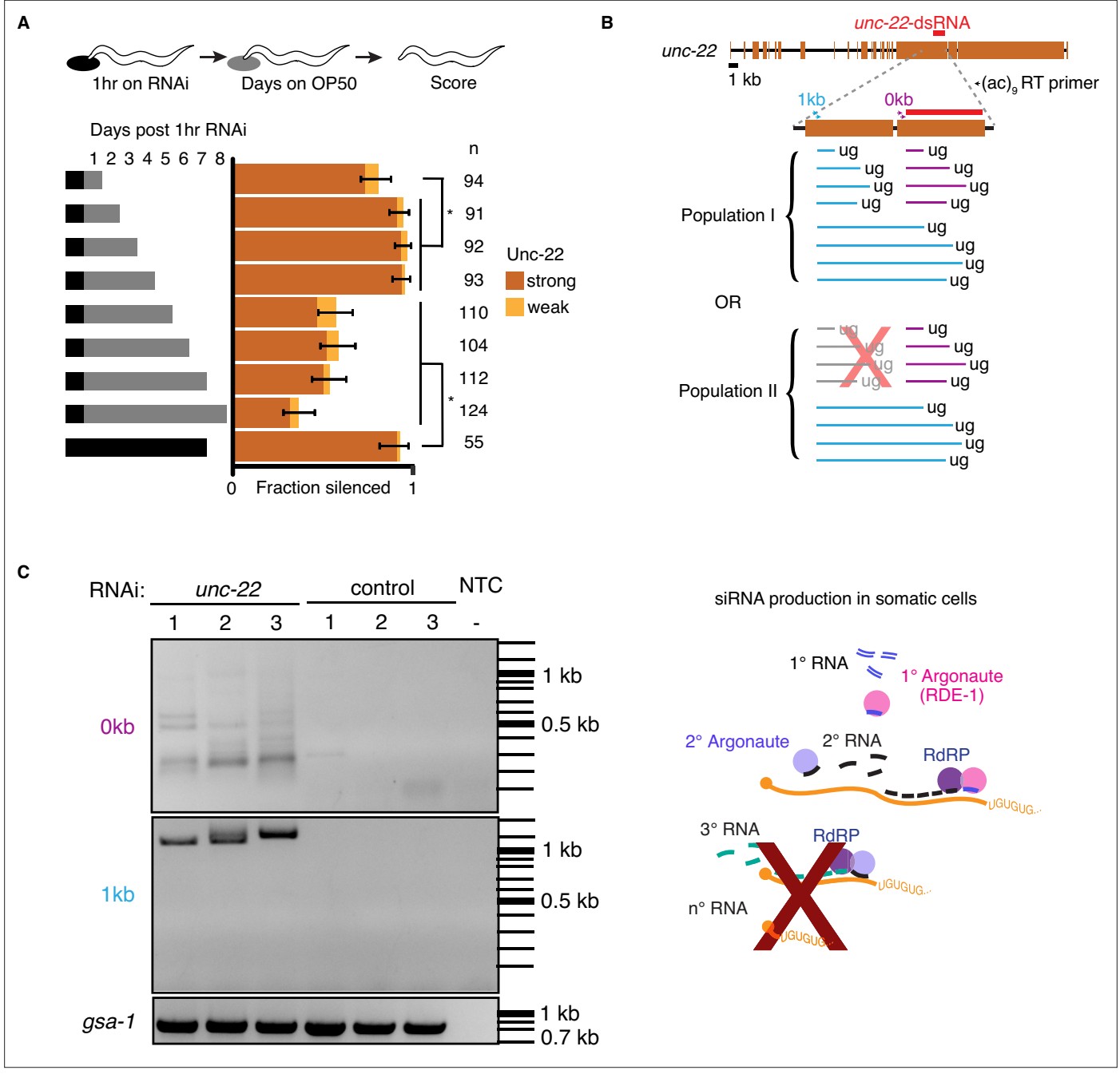

**Figure 6.** Animals recover from a pulse of RNA interference (RNAi) and production of pUG RNAs is restricted despite continuous exposure to double-stranded RNA (dsRNA). (**A**) Response to a pulse of feeding RNAi. (*Top*) Schematic of assay. Animals were exposed to *unc-22* RNAi for 1 hr and then returned to OP50 plates. (*Bottom*) A separate cohort of animals was scored for silencing after each subsequent 24 hr period. Fractions silenced, numbers scored, comparisons, asterisks, and error bars are as in *Figure 1*. A weak Unc-22 defect indicates animals that were nearly completely still except for a slight twitch of the head or of the tail. (**B**) pUG RNA production in response to continuous exposure to *unc-22* dsRNA. (*Top*) Schematic depicting the polymerase chain reaction (PCR) primers used for the RT-PCR to detect pUG RNAs. Two sets of primers (0 kb, purple; 1 kb, blue) positioned 5′ of the *unc-22*-dsRNA (orange), combined with an RT primer that contains nine CA repeats were used. (*Bottom*) Populations of small RNAs that might be detected by pUG-PCR as above. Population I would suggest that amplified 2° short interfering RNAs (siRNAs) in addition to 1° siRNAs are capable of guiding target mRNA cleavage and poly-UG addition. Population II would suggest only 1° siRNAs can initiate pUG RNA production. (**C**) (*Left*) Distribution of DNA amplified from pUG RNAs. Lanes with PCR products amplified from total RNA of animals fed *unc-22* dsRNA (*unc-22*) or L4440 dsRNA (control) isolated from three biological replicates each (1–3), or a no-template control (NTC) with no added RNA (−) are shown. Different bands are detected for each primer set (0 kb, top vs 1 kb, bottom). A gene with a poly-UG sequence encoded in the genome (*gsa-1*) serves as a loading control. (*Right*) Schematic of siRNA production in somatic cells. Successive rounds of amplified small RNAs would map progressively closer to the 5′ end

*Figure 6 continued on next page*

*Figure 6 continued*

of the target transcript. Since we did not detect poly-UG addition upstream of the region homologous to the dsRNA trigger, it is unlikely that 3° siRNAs are being produced.

The online version of this article includes the following source data for figure 6:

**Source data 1.** Excel sheet containing raw data from RNA interference (RNAi) feed depicted in *Figure 6A*.

**Source data 2.** Original, unlabeled image of gel cropped and shown in *Figure 6C*.

**Source data 3.** Complete gel image with annotations to show samples and conditions before being cropped and depicted in *Figure 6C*.

suggests that, during RNAi in somatic cells, the addition of pUG tails is enriched within a zone on target mRNAs that share homology with the dsRNA trigger (pUG zone). This restricted production of pUG RNAs supports the idea that amplification is not perpetual and that mRNA levels can thus recover over time.

## Discussion

Our results suggest that an intersecting network of regulators formed by the intrinsically disordered protein MUT-16, the Maelstrom-domain protein RDE-10, the nuclear Argonaute NRDE-3, and other Argonaute proteins can explain silencing of somatic targets by RNA interference despite stark differences in the genetic requirements for silencing different genes. The requirement for NRDE-3 can be overcome by enhanced dsRNA processing or by changing the *cis*-regulatory sequences of the target gene. However, the combined loss of both MUT-16 and RDE-10 eliminates all detectable silencing and this requirement cannot be overcome by enhancing dsRNA processing. Animals can recover from silencing in non-dividing cells, which supports the turnover of all key RNA intermediates (1° siRNA, 2° siRNA, and pUG RNA). Consistent with the ability to recover from silencing, unlimited rounds of siRNA amplification are curbed by restricting the cleavage and tailing of mRNAs for making pUG RNAs to 'pUG zones' that match the dsRNA sequence (see *Figure 7* for an overview of findings).

### Universal and gene-specific requirements for RNAi

RNAi requires the entry of dsRNA into cells, the processing of dsRNA into small RNAs, recognition of target mRNA, generation of additional small RNAs, and downstream gene silencing mechanisms. The upstream processes of entry, processing, and recognition do not depend on the mRNA being targeted and are thus presumably universal. Consistently, the dsRNA importer SID-1, the endonuclease DCR-1, and the primary Argonaute RDE-1 are required for all RNAi. In contrast, since the mRNA is used as a template to generate the abundant secondary small RNAs in *C. elegans* (*Pak et al., 2012*) or additional dsRNAs in other systems (e.g., in plants; *Sanan-Mishra et al., 2021*), the silencing of different mRNAs could diverge through the selective recruitment of different collections of regulators. In support of this possibility, the two model genes we analyze in this study, *unc-22* and *bli-1*, show stark differences in the requirements of some RNAi factors for silencing (*Figure 1*). While these differences could be attributed to their expression in different tissues, the ability to bypass some requirements (*Figure 3*) argues against this possibility. Specifically, if the requirement for NRDE-3 for silencing *bli-1* (hypodermal gene) but not *unc-22* (muscle gene) is because of the lack of a parallel regulator in the hypodermis but not in the muscle, then enhancing dsRNA processing would be unable to bypass the NRDE-3 requirement (*Figure 3*). Neither the silencing of *dpy-7* nor the silencing of *bli-1p::gfp* requires NRDE-3 despite their expression in the hypodermis (*Figure 2F*, *Figure 5D*). Strikingly,

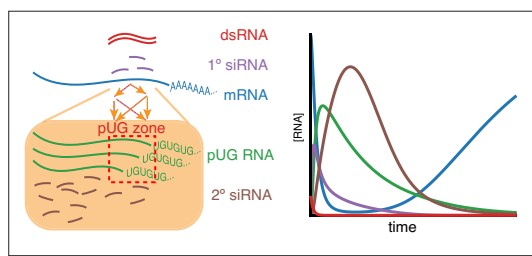

**Figure 7.** Overview of findings. (*Left*) Amplification of small RNAs occurs through an intersecting network of regulators (orange arrows) and in response to somatic RNA interference (RNAi), the addition of poly-UG repeats (green) is restricted to a 'pUG zone' (red) that is homologous to the double-stranded RNA (dsRNA) trigger. (*Right*) In response to a pulse of dsRNA (red), levels of the target mRNA (blue) recover, suggesting that the RNA silencing intermediates (1° short interfering RNAs [siRNAs], purple; pUG RNAs, green; 2° siRNAs, brown) undergo turnover.

changing the 3′*cis*-regulatory sequences of *bli-1* also made its silencing largely independent of NRDE-3 (*Figure 5B*), providing direct evidence for the prior regulation of a gene dictating the genetic requirements for silencing in response to dsRNA. The fact that any two of MUT-16, NRDE-3, and RDE-10 – three structurally and functionally different proteins – are required for *unc-22* silencing suggests that each of these proteins could be contributing to silencing of any RNAi target. Despite this potential use of an intersecting network for silencing all somatic genes, different genes could critically depend on different regulators because of differences in their mRNA metabolism and/or subcellular localization (summarized as threshold differences in *Figure 2*). Intermediate steps that require the participation of mRNA such as the production of 2° siRNA could have complex dependencies, making RNA intermediates poor predictors of silencing efficiency (*Figure 4E*). For example, the subcellular localization of mRNA could increase or decrease its interaction with RdRPs and thus influence the levels of 2° siRNAs made. Future studies that address the dynamics and subcellular localization of target mRNA before RNAi and the subcellular localization of components of the RNAi machinery are required to test these hypotheses.

## Production of 2° siRNAs

Multiple small RNA species of defined lengths and 5′-nt bias have been detected in *C. elegans*. Of these, 22G RNAs (2° siRNAs) are the most abundant and arise from amplification downstream of exposure to dsRNA and in multiple endogenous small RNA pathways (*Gu et al., 2009*). Our results suggest that production of 2° siRNAs in response to dsRNA is eliminated in animals that lack both MUT-16 and RDE-10 (*Figure 3C*). While the precise mechanisms of amplification are unknown, MUT-16 is thought to nucleate perinuclear foci in the germline (*Phillips et al., 2012*; *Uebel et al., 2018*) that recruit many additional components – RDE-2, MUT-7, MUT-14, MUT-15, NYN-1, NYN-2, RDE-8, RDE-3/MUT-2, etc. (*Phillips and Updike, 2022*; *Uebel et al., 2018*). Similar interactions may occur in somatic cells despite the lack of detectable perinuclear foci. The roles of most of these proteins remain obscure, but RDE-8 (*Tsai et al., 2015*) and RDE-3/MUT-2 (*Shukla et al., 2020*) have demonstrated roles in the cleavage and pUGylation of mRNAs, respectively. Yet, the observation of silencing in the absence of MUT-16 that is eliminated upon additional loss of RDE-10 suggests that RDE-10 and associated proteins (e.g., RDE-11 and RDE-12) play an independent role in the amplification of 2° siRNAs (*Yang et al., 2012*; *Zhang et al., 2012*). The subcellular localization of RDE-10 and whether small RNAs that require RDE-10 for production also rely on RDE-8 and RDE-3/MUT-2 as expected for amplification using pUG RNA templates remains to be determined.

Loss of RDE-10 reduces the production of 22G RNAs downstream of exogenous dsRNA and downstream of endogenous small RNAs called 26G RNAs that are 26-nt long and have a 5′G (*Yang et al., 2012*; *Zhang et al., 2012*). Current models for the production of 26G RNAs (*Blumenfeld and Jose, 2016*; *Chaves et al., 2021*) propose that the RdRP RRF-3 transcribes long antisense RNA from internal C nucleotides on template mRNA, the phosphatase PIR-1 converts the 5′ triphosphate of the RdRP product into 5′ mono phosphate, the template is then trimmed by the 3′–5′ exonuclease ERI-1 to generate a blunt-ended dsRNA, which is then cleaved by DCR-1 to generate the mature 26G RNAs that are bound by the Argonaute ERGO-1. While a similar preference by RdRPs can explain the 5′G bias of the downstream 22G RNAs, the mechanism(s) for generating RNA that are precisely 22 nucleotides long remain unclear. This precision could be achieved either through the trimming of template mRNAs into 22-nt long pieces or through the trimming of secondary small RNAs made by RdRPs into 22-nt long pieces. The detection of long pUG RNAs with no detectable shorter pUG RNAs upstream of sequences matching the dsRNA (*Figure 6C*) argues against the 3′ trimming of mRNA templates to generate shorter RNAs that then get pUGylated to become stabilized templates for RdRPs and against pUG RNA generation driven by successive rounds of 22G RNA production in somatic cells. Furthermore, potential 5′ trimming or endonucleolytic cleavage of long pUG RNA to generate a 22-nt template for RdRPs cannot explain the 5′G bias of 22G RNAs. Since Argonautes bind the 5′ end of small RNAs and can associate with RNAs of different lengths (*Ruby et al., 2006*), we suggest a model whereby RDE-10 and downstream Argonautes together play a role in the maturation of 22-nt siRNAs from longer RdRP products.

RDE-10 has a conserved Maelstrom domain that shares homology with the DnaQ-H 3′–5′ exonuclease family (*Zhang et al., 2008*) and the mutation we identified as disrupting silencing by dsRNA (*Figure 1D*) alters a residue located near the highly conserved ECHC zinc-binding motif

(*Figure 1—figure supplement 1C*). Intriguingly, the Maelstrom domain of RDE-10 shares high structural homology with the 3′–5′ exonuclease domain of ERI-1 (*Figure 1—figure supplement 2C*, *left*) but not the exonuclease domain of MUT-7 (*Figure 1—figure supplement 2C*, *right*). ERI-1 can trim single-stranded RNA overhangs in vitro (*Kennedy et al., 2004*) and is required for the production of 26G RNAs (*Duchaine et al., 2006*) and for the maturation of rRNAs (*Gabel and Ruvkun, 2008*). While no 3′–5′ exonuclease activity of RDE-10 or its orthologs has been demonstrated, Maelstrom domain-containing proteins in insects exhibit single-stranded RNA endonuclease activity in vitro (*Matsumoto et al., 2015*). Furthermore, RDE-10 could interact with other parts of the RNA silencing machinery (e.g., the Argonaute ERGO-1 as seen using immunoprecipitation; *Yang et al., 2012*; *Zhang et al., 2012*) to recruit nucleases (e.g., NYN family exonucleases such as ERI-9; *Tsai et al., 2015*) that trim pre-22G RNAs to the 22-nt size preferred by Argonaute proteins. In support of such exonucleolytic trimming in conjunction with Argonaute binding, the 3′–5′ exonuclease SND1 has been shown to trim the 3′ ends of miRNAs bound to AGO1 in *Arabidopsis* (*Chen et al., 2018*). Furthermore, piRNA maturation in *Drosophila* and mice suggests a model where piwi-type Argonautes bind the 5′ end of the pre-piRNA followed by endonucleolytic cutting and exonucleolytic trimming to generate consistently sized mature piRNAs (*Stoyko et al., 2022*). Finally, human ERI1 can trim Ago2-bound micro RNAs to 19-nt (*Sim et al., 2022*).

Therefore, we propose that the production of 22G RNAs in response to the addition of dsRNA occurs as follows: (1) non-processive RdRPs (e.g., RRF-1; *Aoki et al., 2007*) make a heterogenous mixture of short RNAs, (2) 2° Argonautes bind the 5′ end of these pre-secondary siRNA, and (3) RDE-10 and/or associated protein(s) remove excess 3′ sequence to generate 22-nt siRNAs that are effectively retained by the mature siRNA–Argonaute complex. Similar mechanisms could be used to generate other 22G RNAs that are independent of RDE-10 (*Yang et al., 2012*; *Zhang et al., 2012*). Future studies are needed to test each aspect of the model.

## Trade-offs in RNA interference

RNAi is now a widely applied tool for gene silencing in plants, insects, and humans. Like *C. elegans*, plants (*Sanan-Mishra et al., 2021*) and some insects (*Pinzón et al., 2019*) have RdRPs that could be used to make 2° siRNAs, but many other animals, including humans, do not have RdRPs and thus are unlikely to produce 2° siRNAs. However, silencing can fade despite the production of 2° siRNAs (*Figure 6A*), highlighting the importance of dosage for all systems. Two parameters of importance for the acute efficacy of any dsRNA-based drug are the time to knockdown ($\tau_{kd}$ in *Figure 4C*) and duration of knockdown ($t_{kd}$ in *Figure 4C*). The various values of $\tau_{kd}$ that are possible for each $t_{kd}$ (*Figure 4D*) cautions against using a rapid onset of silencing (low $\tau_{kd}$) as the sole indicator of promise during early stages of drug development when long-term effects of a drug are often not evaluated in the interest of expedience. In short, a drug that takes longer to cause an effect could have a more long-lasting effect. Since a dsRNA drug can be synthesized for any target with equal effort, considerations for the choice of target could be worthwhile because differences in RNA metabolism between two targets of equal importance can influence the efficacy of the dsRNA drug in all systems. If two genes are at steady state, then the gene with higher mRNA turnover will be more difficult to knockdown because of higher rates of mRNA production (*Figure 4G*). Similarly, in the absence of a steady state, a gene undergoing upregulation of transcription, splicing, and/or mRNA export during the administration of the drug will be difficult to knockdown (e.g., *Figure 4F*).

In the longer term, a concern for any drug is the development of resistance. When a gene with a high threshold for silencing is targeted, it could rely on multiple regulators that act in parallel to contribute to silencing (e.g., *bli-1* in this study), making resistance through the mutation of any one regulator more likely and necessitating another round of drug development. In contrast, genes with a lower threshold may not require all the regulators for silencing (e.g., *unc-22* in this study), making them ideal targets that remain silenced despite single mutations in many regulators of RNAi (e.g., RDE-10, MUT-16, or NRDE-3 in this study). These trade-offs inform the choice of therapeutic targets and dosage for sustained use of dsRNA-based drugs in agriculture and in human health. Anticipating mechanisms for the development of resistance before widespread use of an RNAi-based drug or treatment will be crucial for avoiding futile cycles of innovation. The ideal drug would require a minimal dose and use multiple intersecting paths to silence the target gene.

# Materials and methods

**Key resources table**

| Reagent type (species) or resource | Designation | Source or reference | Identifiers | Additional information |
|---|---|---|---|---|
| Software, algorithm | Python | https://www.python.org/downloads/release/python-385/ | | |
| Software, algorithm | R | https://cran.r-project.org/bin/macosx/ | | |
| Strains (*C. elegans*) | Please see *Table 1* for complete list | Please see *Table 1* for complete list | Please see *Table 1* for complete list | Please see *Table 1* for complete list |

## Summary

All strains (*Table 1*) were grown at 20°C on Nematode Growth Medium (NGM) plates seeded with OP50 *E. coli* (*Brenner, 1974*). Strains with mutations were generated through a genetic screen after mutagenesis using *N*-ethyl-*N*-nitrosourea (ENU), using standard genetic crosses (*Brenner, 1974*), or using Cas9-mediated genome editing (*Arribere et al., 2014*; *Dokshin et al., 2018*; *Paix et al., 2015*). Mutations induced upon ENU exposure were identified using whole-genome sequencing (Illumina) followed by analyses of the resultant fastq files. Simulations of the RNAi response were used to identify the domain and range of values consistent with experimental data (*Figure 2*) and to explore parameters that support silencing (equilibrium model (*Figure 4—figure supplement 1*) and dynamic model (*Figure 4* and *Figure 4—figure supplement 2*)). Feeding RNAi experiments were performed by exposing worms to bacteria that express dsRNA (*Kamath et al., 2003*; *Timmons and Fire, 1998*) either continuously or for a brief period (*Figure 6A*). Multiple sequence alignment (*Figure 1—figure supplement 2*) was performed using Clustal Omega (*Sievers et al., 2011*) and manually annotated using Illustrator (Adobe). Comparisons of protein structures were performed using AlphaFold predictions (*Jumper et al., 2021*; *Varadi et al., 2022*), pairwise alignment on Protein Data Bank (*Zhang and Skolnick, 2005*), and the PyMOL Molecular Graphics System (v. 2.4.1 Schrödinger, LLC). Levels of *rde-4* mRNA (*Figure 3—figure supplement 1*) and pUG RNA (*Figure 6C*) were measured using reverse-transcription followed by polymerase chain reaction (RT-PCR). Transgenic strains that express *rde-1(+)* and *rde-4(+)* in specific tissues were generated using Mos1-mediated single-copy insertion (MosSCI, *Frøkjær-Jensen et al., 2012*). Oligonucleotides used are in *Table 2*. Exact p-values and additional details for each experiment are in *Table 3*. All code used (R, Python, and Shell) is available on GitHub (copy archived at *Knudsen et al., 2024*).

## Strains and oligonucleotides used

All strains (listed in *Table 1*) were cultured on NGM plates seeded with 100 µl of OP50 *E. coli* at 20°C and strains made through mating were generated using standard methods (*Brenner, 1974*). Oligonucleotides used are in *Table 2*. Strains generated using MosSCI (*Frøkjær-Jensen et al., 2012*) of *rde-4* or *rde-1* rescues in the germline (as in *Marré et al., 2016*) or of *rde-4* rescues in the hypodermis (*Raman et al., 2017*) were used in this study.

## Genetic screen

This screen was performed by mutagenizing a strain (AMJ174) with the transgene *T* (*oxSi487[mex-5p::mCherry::H2B::tbb-2 3'UTR::gpd-2 operon::GFP::H2B::cye-1 3'UTR +unc-119(+)]*, *Devanapally et al., 2021*) silenced for >200 generations after introducing a mutation in *lin-2(jam30)* (sgRNA (P1), primers (P2, P3, P4)) using Cas9-mediated genome editing of AMJ844 (*iT*; *dpy-2(e8)*, *Devanapally et al., 2021*) while correcting the *dpy-2(e8)* mutation to wild type (creating *dpy-2(jam29)*; sgRNA (P5), primers (P6, P7, P8)). The *lin-2* mutation limits brood size (*Ferguson and Horvitz, 1985*) and facilitates screening. Near-starved animals (P0) of all life stages were mutagenized using 1 mM ENU (Toronto Research Chemicals) for 4–6 hr. Mutagenized animals were washed four times with wash buffer (0.01% Triton X-100 in M9) and two to three adult animals were placed on NG plates seeded with OP50. Over the next 3 weeks, F1, F2, and F3 progeny were screened to isolate mutants that show mCherry fluorescence. These animals were singled out (up to 7 animals from each P0 plate) and tested for the

**Table 1.** Strains used in this study.

| Strain name | Genotype |
| --- | --- |
| N2 | Wild type |
| AMJ174 | oxSi487[mex-5p::mCherry::H2B::tbb-2 3'UTR::gpd-2 operon::GFP::H2B::cye-1 3'UTR + unc-119(+)] dpy-2(jam29) II; unc-119(ed3)? III; lin-2(jam30) X |
| AMJ183 | rde-4(ne301) III; nrde-3(tm1116) X |
| AMJ285 | jamSi1 [mex-5p::rde-4(+)] II; rde-4(ne301) III |
| AMJ345 | jamSi2 [mex-5p::rde-1(+)] II; rde-1(ne219) V |
| AMJ422 | jamSi6 [nas-9p::rde-4(+)] II; unc-119(ed) III |
| AMJ489 | nrde-3(tm1116) X; eri-1(mg366) IV |
| AMJ565 AMJ611 | jamSi6 [Pnas-9::rde-4(+)::rde-4 3'UTR] II; unc-119(ed3) III rde-4(ne301) III jamSi6 [nas-9p::rde-4(+)::rde-4 3'UTR] II; rde-4(ne301) III; nrde-3(tm1116) X |
| AMJ1023 | mut-16(jam138) I; oxSi487 dpy-2(jam29) II; unc-119(ed3)? III; lin-2(jam30) X |
| AMJ1025 | mut-16(jam139) rde-10(jam248) I; oxSi487 dpy-2(jam29) II; unc-119(ed3)? III; lin-2(jam30) X |
| AMJ1035 | mut-16(jam140) I; oxSi487 dpy-2(jam29) II; unc-119(ed3)? III; lin-2(jam30) X |
| AMJ1042 | mut-16(jam141) I; oxSi487 dpy-2(jam29) II; unc-119(ed3)? III; lin-2(jam30) X |
| AMJ1091 | mut-16(jam247) I; oxSi487 dpy-2(jam29) II; unc-119(ed3)? III; lin-2(jam30) X |
| AMJ1195 | jamSi59 [Pmex-5::gfp::cye-1 3'UTR + unc-119(+)] II; unc-119(ed3) III |
| AMJ1397 | mut-16(jam148) I |
| AMJ1470 | mut-16(jam148) rde-10(jam196) I |
| AMJ1489 | rde-10(jam206) I |
| AMJ1510 | nrde-3(jam205) X |
| AMJ1545 | mut-16(jam148) I; nrde-3(jam205) X |
| AMJ1568 | rde-10(jam206) I; nrde-3(jam205) X |
| AMJ1611 | mut-16(jam240) rde-10(jam206) I |
| AMJ1614 | rde-10(jam243) I |
| AMJ1621 | eri-1(mg366) IV; nrde-3(jam205) X |
| AMJ1622 | rde-10(jam206) I; eri-1(mg366) IV |
| AMJ1623 | rde-10(jam206) I; eri-1(mg366) IV |
| AMJ1624 | rde-10(jam206) I; eri-1(mg366) IV; nrde-3(jam205) X |
| AMJ1625 | rde-10(jam206) I; eri-1(mg366) IV; nrde-3(jam205) X |
| AMJ1631 | mut-16(jam148) I; eri-1(mg366) IV; nrde-3(jam205) X |
| AMJ1632 | mut-16(jam148) I; eri-1(mg366) IV; nrde-3(jam205) X |
| AMJ1657 | mut-16(jam240) rde-10(jam206) I; eri-1(jam260) IV |
| AMJ1658 | mut-16(jam240) rde-10(jam206) I; eri-1(jam261) IV |
| AMJ1659 | mut-16(jam240) rde-10(jam206) I; eri-1(jam262) IV |
| AMJ1660 | eri-1(jam263) IV |
| AMJ1661 | eri-1(jam264) IV |
| AMJ1672 | mut-16(jam265) I; eri-1(jam263) IV |
| AMJ1673 | mut-16(jam266) I; eri-1(jam264) IV |
| AMJ1674 | mut-16(jam267) I; eri-1(jam264) IV |
| AMJ1675 | mut-16(jam268) I |
| AMJ1709 | jam297[bli-1p::gfp::cye-1 3'utr + unc-119(+)] II; unc-119(ed3) III |
| AMJ1721 | mut-16(jam148) I; jamSi6 [nas-9p::rde-4(+)] II; unc-119(ed)? III |

*Table 1 continued on next page*

*Table 1 continued*

| Strain name | Genotype |
| --- | --- |
| AMJ1722 | *mut-16(jam148) I; jamSi6 II; unc-119(ed)? III; eri-1(jam263) IV* |
| AMJ1723 | *rde-10(jam206) I; jamSi6 II; unc-119(ed)? III* |
| AMJ1724 | *rde-10(jam206) I; jamSi6 II; unc-119(ed)? III; eri-1(mg366) IV* |
| AMJ1725 | *rde-10(jam206) I; jam297[bli-1p::gfp::cye-1 3'utr + unc-119(+)] II; unc-119(ed3)? III* |
| AMJ1726 | *jam297[bli-1p::gfp::cye-1 3'utr + unc-119(+)] II; unc-119(ed3)? III; nrde-3(jam205) X* |
| AMJ1727 | *mut-16(jam148) I; jam297[bli-1p::gfp::cye-1 3'utr + unc-119(+)] II; unc-119(ed3)? III* |
| AMJ1728 | *mut-16(jam148) rde-10(jam196) I; jam297[bli-1p::gfp::cye-1 3'utr + unc-119(+)] II; unc-119(ed3)? III* |
| AMJ1730 | *unc-22(jam300) IV* |
| AMJ1731 | *unc-22(jam301) IV* |
| AMJ1754 | *bli-1(jam307) II* |
| AMJ1755 | *bli-1(jam308) II* |
| AMJ1757 | *unc-22(jam300) IV; nrde-3(jam205) X* |
| AMJ1758 | *bli-1(jam307) II; nrde-3(jam205) X* |
| AMJ1771 | *bli-1(jam308) II; nrde-3(jam205) X* |
| EG6787 | *oxSi487 II; unc-119(ed3) III* |
| GR1373 | *eri-1(mg366) IV* |
| WM27 | *rde-1(ne219) V* |
| WM49 | *rde-4(ne301) III* |
| WM156 | *nrde-3(tm1116) X* |

persistence of expression in descendants. Of the 15 fertile mutants isolated using this primary screen, five with mutations in *mut-16* were analyzed in this study.

## Whole-genome sequencing

Libraries were prepared using TruSeq DNA Library Prep kits (Illumina) and samples were sequenced at Omega Biosciences. The fastq files obtained after Illumina sequencing (1× PE 150 b, Omega Biosciences) were analyzed to identify candidate mutations responsible for the observed defects in the sequenced strains. For each strain, sequences were trimmed using cutadapt (v. 3.5), mapped to the *C. elegans* genome (WBcel235/ce11) using bowtie2 (v. 2.4.2), sorted using samtools (v. 1.11), and the resulting.bam file was analyzed to call variants using snpEff (v. 5.0e). The variants classified as 'HIGH' or 'MODERATE' in the.ann.vcf file for each strain that were not shared by any two or more strains were culled as new mutations caused by mutagenesis in each strain. These new mutations in each strain were compared with those of all other strains (in silico complementation) using a custom script to identify sets of strains with different mutations in the same genes. Specific details for each step are provided within the scripts '1_fastq_to_sorted_bam.sh', '2_sorted_bam_to_mutated_genes.sh', '3_in_silico_complementation.sh' available on GitHub (copy archived at *Knudsen et al., 2024*). Raw fastq files for the strains analyzed in this study (AMJ1023, AMJ1025, AMJ1035, AMJ1042, and AMJ1091) have been submitted to SRA (PRJNA928750).

## Modeling and simulation

The RNAi response was explored using three models of increasing complexity: (1) a single-network model of protein factors with branching pathways for RNA amplification and subsequent gene silencing (*Figure 2*); (2) an equilibrium model for the dependence of mRNA and pre-mRNA on small RNAs and other RNA intermediates (*Figure 4—figure supplement 1*); and (3) a dynamic model using ordinary differential equations for the dependence of mRNA and pre-mRNA on small RNAs and other RNA intermediates (*Figure 4* and *Figure 4—figure supplement 2*). Simulations of single network and

**Table 2.** Oligonucleotides used in this study.

| Primer | Sequence |
| --- | --- |
| P1 | atttaggtgacactatagaaatgctcagagatgctcggttttagagctagaaatagcaag |
| P2 | tcactttcttcgtgcgttcc |
| P3 | ggagaaccactcccagaatg |
| P4 | aatcaatcggctgtccacac |
| P5 | atttaggtgacactatagctggatcacctgggaatccgttttagagctagaaatagcaag |
| P6 | aatcgcaaacgagtgggtac |
| P7 | cgggctagatcataatgagg |
| P8 | ggaccacgtggagttccaggacatccaggtttttccaggtgacccaggagagtatggaatt |
| P9 | gaatatttttcgaaaatata |
| P10 | cggcacatgcgaatattttccgaaaatagaaggatattcttcaactcgatccagaaaaac |
| P11 | gctaccataggcaccgcatg |
| P12 | cacttgaacttcaatacggcaagatgagaatgactggaaaccgtaccgcatgcggtgcctatggtagcggagcttcacatggcttcagaccaacagccta |
| P13 | cacaaacgccaggaaaggaag |
| P14 | catttctgcgttgttgtggacc |
| P15 | gttgtaacggatatctctgc |
| P16 | aagattgaatgttgtaacgaatatttcagcaggatacgatgaaagcttattgattgatgg |
| P17 | ccgaaatccagatgagttcc |
| P18 | gcatctggataaaaccaagc |
| P19 | ccgatacaatcagaatgatc |
| P20 | agcaaggccaccgatacaatcagaatgat**t**aggcagacaaggatattatgacaagatatt |
| P21 | ggcattcgagccaataatgc |
| P22 | cgttgtgctcggcaacttct |
| P23 | acaccacgtacaaatgtttg |
| P24 | tgcgtcatccacaccacgtacaaacgtttagggcactgcaaaaaagccatccagccaaca |
| P25 | gactgtgctgacgctgtttt |
| P26 | ctcccagtggctttcgtttt |
| P27 | tgctgctccatatttccgag |
| P28 | gaaacagtcgatgctgctccatatttccgataggatcttcaacggctgtacacatggatg |
| P29 | cctatgtccgacctgtcaga |
| P30 | caattccggatttctgaagag |
| P31 | cagacctcacgatatgtggaaa |
| P32 | ggaacatatggggcattcg |
| P33 | caactttgtatagaaaagtt |
| P34 | acaagtttgtacaaaaaagc |
| P35 | gattacgccaagctatcaactttgtatagaaaagttgcctaccaaagtagaaattcc |
| P36 | acaactccagtgaaaagttcttctcctttactcatgatgaggttagatcacacta |
| P37 | tttcgctgtcctgtcacactc |
| P38 | tacgcggtaagacccaaatg |
| P39 | gaacgcgtcgaggtgatagc |
| P40 | ataaggagttccacgcccag |

*Table 2 continued on next page*

*Table 2 continued*

| Primer | Sequence |
|---|---|
| P41 | ctagtgagtcgtattataagtg |
| P42 | tgaagacgacgagccacttg |
| P43 | ctagaaacttctcataatag |
| P44 | ggatacgagagaagccaaat |
| P45 | cttttacaggaaccactattatgagaagtttctagtttaatcatcctgccaccaccactt |
| P46 | ccttatcttctgcggttttcccaactctccgcttcttccaaacatttctcagtcaacag |
| P47 | catacagaaggagaaatcgc |
| P48 | gttgtagtacagtgtcgcat |
| P49 | gcgtcccaattcttgaatca |
| P50 | ggtggcaggatgattagaca |
| P51 | aattctcactcaaaatttgc |
| P52 | tgcaaaatatgcggcagctcttctccttgtctaataactaaaaaaaacttctagtctaac |
| P53 | tgtctttcaaattctcactcaaaatttgctggtatcgatttggcttctctcgtatcc |
| P54 | gacgacgacggcatctatgt |
| P55 | gctatggctgttctcatggcggcgtcgccatattctacttcacacacacacacacaca |
| P56 | gctatggctgttctcatggc |
| P57 | gagttctacgatcacattct |
| P58 | tgctccgtggagcaactcgc |
| P59 | gagcacactattctgtgcat |
| P60 | ggcgtcgccatattctactt |
| P61 | cacttgctggaaagacaagg |
| P62 | cgcaagcatgctggtttgta |
| P63 | gcattccatctgcaatgcga |
| P64 | gccgatttacaagcacactg |
| P65 | tcgtcttcggcagttgcttc |
| P66 | gcaaagaatcttgcagcatgg |
| P67 | tcttcagtctgggtgtgttc |
| P68 | gacgagcaaatgctcaac |
| P69 | ttcggtgaactccatctcg |

exploration of equilibrium model were conducted in R (v. 3.6.3). Simulations of the dynamic model were conducted in Python (v. 3.8.5) and in R (v. 4.1.0).

## Intersecting network

Random numbers from 0 to 2 were selected for each of the assigned variables (Nm, Or, *Nm*, *Or*) and parameter sets that satisfy experimental constraints were plotted. Specific details are provided within the script '2022_6_13_RNAi_in_Celegans_linear_modified.R' available on GitHub (copy archived at **Knudsen et al., 2024**).

## Equilibrium model

This model for RNAi interference assumes that all reactions have reached equilibrium. Additional assumptions include (1) 1° siRNAs, then pUG RNAs, then 2° siRNAs are made sequentially, (2) no 3° siRNAs are produced for these somatic targets (supported by **Pak et al., 2012**), (3) there is no recycling of full-length mRNA or full-length pre-mRNA after small RNA binding, that is, multiple rounds

**Table 3.** Summary of statistics.

Comparisons with a p-value of less than 0.05 are denoted with "*" and "†" indicates a p-value of less than 0.05 when compared to wild type and the *rde-4(ne301)* mutant (in *Figure 3D*).

| Figure | Comparison | Total *n* | Silenced *n* | p-value | Strains | Notes |
|---|---|---|---|---|---|---|
| 1B *bli-1* | wild type vs *mut-16(jam138)* | 757, 229 | 391, 0 | <0.00001, * | EG6787, AMJ1023 | Pooled EG6787 from separate experiments |
| 1B *bli-1* | wild type vs *mut-16(jam140)* | 757, 277 | 391, 0 | <0.00001, * | EG6787, AMJ1035 | Pooled EG6787 from separate experiments |
| 1B *bli-1* | wild type vs mut-16(jam141) | 757, 124 | 391, 0 | <0.00001, * | EG6787, AMJ1042 | Pooled EG6787 from separate experiments |
| 1B *bli-1* | wild type vs mut-16(jam247) | 757, 446 | 391, 1 | <0.00001, * | EG6787, AMJ1091 | Pooled EG6787 from separate experiments |
| 1B *bli-1* | wild type vs mut-16(jam139) rde-10(jam248) | 757, 412 | 391, 1 | <0.00001, * | EG6787, AMJ1025 | Pooled EG6787 from separate experiments |
| 1B *unc-22* | wild type vs mut-16(jam139) rde-10(jam248) | 309, 173 | 282, 0 | <0.00001, * | EG6787, AMJ1025 | Pooled EG6787 from separate experiments |
| 1B *unc-22* | mut-16(jam138) vs mut-16(jam139) rde-10(jam248) | 180, 173 | 111, 0 | <0.00001, * | AMJ1023, AMJ1025 | Pooled EG6787 from separate experiments |
| 1B *unc-22* | mut-16(jam140) vs mut-16(jam139) rde-10(jam248) | 91, 173 | 50, 0 | <0.00001, * | AMJ1035, AMJ1025 | Pooled EG6787 from separate experiments |
| 1B *unc-22* | mut-16(jam141) vs mut-16(jam139) rde-10(jam248) | 267, 173 | 208, 0 | <0.00001, * | AMJ1042, AMJ1025 | Pooled EG6787 from separate experiments |
| 1B *unc-22* | mut-16(jam247) vs mut-16(jam139) rde-10(jam248) | 100, 173 | 84, 0 | <0.00001, * | AMJ1091, AMJ1025 | Pooled EG6787 from separate experiments |
| 1D *bli-1* | wild type vs mut-16(jam148) | 202, 126 | 159, 0 | <0.00001, * | N2, AMJ1397 | |
| 1D *bli-1* | wild type vs mut-16(jam148) rde-10(jam196) | 202, 209 | 159, 0 | <0.00001, * | N2, AMJ1470 | |
| 1D *bli-1* | wild type vs rde-10(jam206) | 202, 209 | 159, 0 | <0.00001, * | N2, AMJ1489 | |
| 1D *unc-22* | wild type vs mut-16(jam148) | 206, 121 | 204, 119 | 0.58802, ns | N2, AMJ1397 | |
| 1D *unc-22* | wild type vs mut-16(jam148) rde-10(jam196) | 206, 146 | 204, 0 | <0.00001, * | N2, AMJ1470 | |
| 1D *unc-22* | wild type vs rde-10(jam206) | 206, 190 | 204, 185 | 0.21023, ns | N2, AMJ1489 | |
| 2B *bli-1* | wild type vs nrde-3(jam205) | 295, 274 | 172, 16 | <0.00001, * | N2, AMJ1510 | |
| 2B *bli-1* | wild type vs rde-10(jam206); nrde-3(jam205) | 295, 219 | 172, 0 | <0.00001, * | N2, AMJ1568 | |
| 2B *bli-1* | wild type vs mut-16(jam148); nrde-3(jam205) | 295, 121 | 172, 0 | <0.00001, * | N2, AMJ1545 | |
| 2B *unc-22* | wild type vs nrde-3(jam205) | 129, 110 | 127, 108 | 0.87221, ns | N2, AMJ1510 | |
| 2B *unc-22* | wild type vs rde-10(jam206); nrde-3(jam205) | 129, 111 | 127, 0 | <0.00001, * | N2, AMJ1568 | |
| 2B *unc-22* | wild type vs mut-16(jam148); nrde-3(jam205) | 129, 101 | 127, 0 | <0.00001, * | N2, AMJ1545 | |
| 2E | wild type vs mut-16(jam148) | 183, 107 | 179, 5 | <0.00001, * | N2, AMJ1397 | |
| 2E | wild type vs rde-10(jam206) | 183, 116 | 179, 16 | <0.00001, * | N2, AMJ1489 | |

*Table 3 continued on next page*

*Table 3 continued*

| Figure | Comparison | Total *n* | Silenced *n* | p-value | Strains | Notes |
|---|---|---|---|---|---|---|
| 2E | wild type vs nrde-3(jam205) | 183, 105 | 179, 72 | <0.00001, * | N2, AMJ1510 | |
| 2E | mut-16(jam148) vs rde-10(jam206) | 107, 116 | 5, 16 | 0.0190241, * | AMJ1397, AMJ1489 | |
| 2E | rde-10(jam206) vs nrde-3(jam205) | 116, 105 | 16, 72 | <0.00001, * | AMJ1489, AMJ1510 | |
| 2F *unc-54* | wild type vs mut-16(jam148) | 148, 124 | 142, 2 | <0.00001, * | N2, AMJ1397 | |
| 2F *unc-54* | wild type vs rde-10(jam206) | 148, 128 | 142, 2 | <0.00001, * | N2, AMJ1489 | |
| 2F *unc-54* | wild type vs nrde-3(jam205) | 148, 263 | 142, 49 | <0.00001, * | N2, AMJ1510 | Fed on a different day with similar N2 silencing 164/171 animals showing Unc-54 |
| 2F *unc-54* | wild type vs mut-16(jam148) rde-10(jam196) | 148, 159 | 142, 0 | <0.00001, * | N2, AMJ1470 | |
| 2F *unc-54* | wild type vs mut-16(jam148); nrde-3(jam205) | 148, 166 | 142, 0 | <0.00001, * | N2, AMJ1545 | Fed on a different day with similar N2 silencing 164/171 animals showing Unc-54 |
| 2F *unc-54* | wild type vs rde-10(jam206); nrde-3(jam205) | 148, 206 | 142, 0 | <0.00001, * | N2, AMJ1568 | Fed on a different day with similar N2 silencing 164/171 animals showing Unc-54 |
| 2F *unc-54* | nrde-3(jam205) vs mut-16(jam148) | 263, 124 | 49, 2 | <0.00001, * | AMJ1510, AMJ1397 | |
| 2F *unc-54* | nrde-3(jam205) vs rde-10(jam206) | 263, 128 | 49, 2 | <0.00001, * | AMJ1510, AMJ1489 | |
| 2F *dpy-7* | wild type vs mut-16(jam148) | 322, 130 | 279, 34 | <0.00001, * | N2, AMJ1397 | |
| 2F *dpy-7* | wild type vs rde-10(jam206) | 322, 159 | 279, 2 | <0.00001, * | N2, AMJ1489 | |
| 2F *dpy-7* | wild type vs nrde-3(jam205) | 322, 126 | 279, 3 | <0.00001, * | N2, AMJ1510 | |
| 2F *dpy-7* | wild type vs mut-16(jam148) rde-10(jam196) | 322, 184 | 279, 0 | <0.00001, * | N2, AMJ1470 | |
| 2F *dpy-7* | wild type vs mut-16(jam148); nrde-3(jam205) | 322, 99 | 279, 0 | <0.00001, * | N2, AMJ1545 | |
| 2F *dpy-7* | wild type vs rde-10(jam206); nrde-3(jam205) | 322, 180 | 279, 0 | <0.00001, * | N2, AMJ1568 | |
| 2F *dpy-7* | mut-16(jam148) vs rde-10(jam206) | 130, 159 | 34, 2 | <0.00001, * | AMJ1397, AMJ1489 | |
| 2F *dpy-7* | mut-16(jam148) vs nrde-3(jam205) | 130, 126 | 34, 6 | <0.00001, * | AMJ1397, AMJ1510 | |
| 2F | wild type vs mut-16(jam148) | 183, 107 | 179, 5 | <0.00001, * | N2, AMJ1397 | |
| 2F | wild type vs rde-10(jam206) | 183, 116 | 179, 16 | <0.00001, * | N2, AMJ1489 | |
| 2F | wild type vs nrde-3(jam205) | 183, 105 | 179, 72 | <0.00001, * | N2, AMJ1510 | |
| 2F | mut-16(jam148) vs rde-10(jam206) | 107, 116 | 5, 16 | 0.0190241, * | AMJ1397, AMJ1489 | |
| 2F | rde-10(jam206) vs nrde-3(jam205) | 116, 105 | 16, 72 | <0.00001, * | AMJ1489, AMJ1510 | |

*Table 3 continued on next page*

*Table 3 continued*

| Figure | Comparison | Total *n* | Silenced *n* | p-value | Strains | Notes |
|---|---|---|---|---|---|---|
| 3B | wild type vs eri-1(mg366) | 200, 200 | 134, 145 | 0.231163, ns | N2, GR1373 | N2 from a second experiment showed comparable values (219/304), GR1373 from a second experiment showed comparable values (169/213) |
| 3B | nrde-3(jam205) vs eri-1(mg366); nrde-3(tm1116) | 143, 200 | 3, 167 | 0.000132, * | N2, AMJ489 | |
| 3B | mut-16(148/268) vs mut-16(jam265-7); eri-1(jam263) | 172, 354 | 0, 0 | >0.5, ns | AMJ1397, AMJ1675, AMJ1672-4 | AMJ1397 (0/110) and AMJ1675 (0/123) were pooled; AMJ1672 (0/130), AMJ1673 (0/88), and AMJ1674 (0/111) were pooled |
| 3B | rde-10(jam206) vs rde-10(jam206); eri-1(mg366) | 233, 329 | 0, 0 | >0.5, ns | AMJ1489, AMJ1622, AMJ1623 | AMJ1622 (0/171) and AMJ1623 (0/183) were pooled |
| 3B | nrde-3(jam205) vs eri-1(mg366); nrde-3(jam205) | 143, 169 | 3, 130 | <0.00001, * | AMJ1510, AMJ1621 | |
| 3C | mut-16(jam148/240) rde-10(jam196/206) vs mut-16(jam240) rde-10(jam 206); eri-1(jam260-2) | 149, 191 | 1, 10 | 0.01826, * | AMJ1470, AMJ1611, AMJ1657-9 | AMJ1470 (0/32) and AMJ1661 (1/116) were pooled; AMJ1657 (9/75), AMJ1658 (0/56), and AMJ1659 (1/50) were pooled |
| 3C | rde-10(jam206); nrde-3(jam205) vs rde-10(jam206); eri-1(mg366); nrde-3(jam205) | 113, 265 | 6, 18 | 0.58837, ns | AMJ1568, AMJ1624, AMJ1625 | AMJ1624 (4/135) and AMJ1625 (14/112) were pooled |
| 3C | mut-16(jam148); nrde-3(jam205) vs mut-16(jam148); eri-1(mg366); nrde-3(jam205) | 28, 110 | 4, 1 | 0.00072, * | AMJ1545, AMJ1631, AMJ1632 | AMJ1631 (1/52) and AMJ1632 (0/57) were pooled |
| 3D *bli-1* | wild type vs rde-1(ne219) | 50, 37 | 41, 0 | <0.00001, * | N2, WM27 | N2 data from *rde-4(−)* experiment; N2 data from *rde-1(−)* experiment (not shown) is comparable with 41/50 silenced |
| 3D *bli-1* | wild type vs rde-4(ne301) | 40, 50 | 40, 0 | <0.00001, * | N2, WM49 | N2 data from *rde-4(−)* experiment; N2 data from *rde-1(−)* experiment (not shown) is comparable with 41/50 silenced |
| 3D *bli-1* | wild type vs Si[mex-5p::rde-1(+)]/+; rde-1(ne219) | 50, 33 | 41, 0 | <0.00001, * | N2, AMJ345 | N2 data from *rde-4(−)* experiment; N2 data from *rde-1(−)* experiment (not shown) is comparable with 41/50 silenced |
| 3D *bli-1* | wild type vs Si[mex-5p::rde-4(+)]/+; rde-4(ne301) | 40, 41 | 40, 41 | >0.5, ns | N2, AMJ285 | N2 data from *rde-4(−)* experiment; N2 data from *rde-1(−)* experiment (not shown) is comparable with 41/50 silenced |
| 3D *unc-22* | wild type vs rde-1(ne219) | 25, 25 | 25, 0 | <0.00001, * | N2, WM27 | |
| 3D *unc-22* | wild type vs rde-4(ne301) | 25, 25 | 25, 0 | <0.00001, * | N2, WM49 | |
| 3D *unc-22* | wild type vs Si[mex-5p::rde-1(+)]/+; rde-1(ne219) | 25, 24 | 25, 2 | <0.00001, * | N2, AMJ345 | |
| 3D *unc-22* | wild type vs Si[mex-5p::rde-4(+)]/+; rde-4(ne301) | 25, 24 | 25, 16 | 0.001600, † | N2, AMJ285 | † indicates statistical significance vs wild type and vs *rde-4(ne301)* mutant |
| 3D *unc-22* | rde-4(ne301) vs Si[mex-5p::rde-4(+)]/+; rde-4(ne301) | 25, 24 | 0, 16 | <0.00001, † | WM49, AMJ285 | † indicates statistical significance vs wild type and vs *rde-4(ne301)* mutant |
| 3E | wild type vs rde-4(ne301); nrde-3(tm1116) | 150, 200 | 142, 0 | <0.00001, * | N2, AMJ183 | |
| 3E | wild type vs rde-4(ne301); nrde-3(tm1116); Si[nas-9p::rde-4(+)] | 150, 150 | 142, 98 | <0.00001, † | N2, AMJ611 | † indicates statistical significance vs wild type and vs *rde-4(ne301); nrde-3(tm1116)* mutant |

*Table 3 continued on next page*

*Table 3 continued*

| Figure | Comparison | Total n | Silenced n | p-value | Strains | Notes |
|---|---|---|---|---|---|---|
| 3E | rde-4(ne301); nrde-3(tm1116); vs rde-4(ne301); nrde-3(tm1116); Si[nas-9p::rde-4(+)] | 200, 150 | 0, 98 | <0.00001, † | AMJ183, AMJ611 | † indicates statistical significance vs wild type and vs *rde-4(ne301); nrde-3(tm1116)* mutant |
| 3F | wild type vs eri-1(mg366); nrde-3(jam205) | 282, 140 | 85, 69 | 0.0001199, * | N2, AMJ1621 | |
| 3F | wild type vs rde-10(jam206); eri-1(mg366) | 282, 279 | 85, 0 | <0.00001, * | N2, AMJ1622 | |
| 3F | wild type v Si[nas-9p::rde-4(+)]; rde-10(jam206) | 282, 180 | 85, 6 | <0.00001, * | N2, AMJ1723 | |
| 3F | wild type v Si[nas-9p::rde-4(+)]; rde-10(jam206); eri-1(mg366) | 282, 202 | 85, 0 | <0.00001, * | N2, AMJ1724 | |
| 3F | wild type vs mut-16(jam148); eri-1(mg366) | 282, 70 | 85, 0 | <0.00001, * | N2, AMJ1672 | |
| 3F | wild type v Si[nas-9p::rde-4(+)]; mut-16(jam148) | 282, 152 | 85, 0 | <0.00001, * | N2, AMJ1721 | |
| 3F | wild type v Si[nas-9p::rde-4(+)]; mut-16(jam148); eri-1(mg366) | 282, 187 | 85, 0 | <0.00001, * | N2, AMJ1722 | |
| 5B | wild type vs bli-1(jam307) | 488, 443 | 390, 425 | <0.00001, * | N2, AMJ1754 | |
| 5B | wild type vs bli-1(jam308) | 488, 245 | 390, 185 | 0.170996, ns | N2, AMJ1755 | |
| 5B | wild type vs nrde-3(jam205) | 488, 377 | 390, 111 | <0.00001, * | N2, AMJ1510 | |
| 5B | bli-1(jam307) vs bli-1(jam307); nrde-3(jam205) | 443, 459 | 425, 313 | <0.00001, * | AMJ1754, AMJ1758 | |
| 5B | nrde-3(jam205) vs bli-1(jam307); nrde-3(jam205) | 377, 459 | 111, 313 | <0.00001, * | AMJ1510, AMJ1758 | |
| 5B | bli-1(jam308) vs bli-1(jam308); nrde-3(jam205) | 245, 117 | 185,38 | <0.00001, * | AMJ1755, AMJ1771 | |
| 5B | wild type vs bli-1(jam307) | 488, 443 | 390, 425 | <0.00001, * | N2, AMJ1754 | |
| 5D | *bli-1p::gfp* vs mut-16(jam148); *bli-1p::gfp* | 21, 25 | 21, 6 | <0.00001, * | AMJ1709, AMJ1727 | |
| 5D | *bli-1p::gfp* vs rde-10(jam206); *bli-1p::gfp* | 21, 25, | 21, 9 | <0.00001, * | AMJ1709, AMJ1725 | |
| 5D | *bli-1p::gfp* vs nrde-3(jam205); *bli-1p::gfp* | 21, 25 | 21, 23 | 0.18508, ns | AMJ1709, AMJ1726 | |
| 5D | *bli-1p::gfp* vs mut-16(jam148)rde-10(jam196); *bli-1p::gfp* | 21, 20 | 21, 0 | <0.00001, * | AMJ1709, AMJ1728 | |
| 6A | 1 vs 2 days post RNAi | 94, 91 | 76, 86 | 0.00489, * | N2 | |
| 6A | 1 vs 3 days post RNAi | 94, 92 | 76, 89 | 0.00062, * | N2 | |
| 6A | 1 vs 4 days post RNAi | 94, 93 | 76, 89 | 0.001627, * | N2 | |
| 6A | 5 days post RNAi vs 7 days on RNAi | 110, 55 | 63, 51 | <0.00001, * | N2 | |
| 6A | 6 days post RNAi vs 7 days on RNAi | 104, 55 | 61, 51 | <0.00001, * | N2 | |
| 6A | 7 days post RNAi vs 7 days on RNAi | 112, 55 | 60, 51 | <0.00001, * | N2 | |

*Table 3 continued on next page*

*Table 3 continued*

| Figure | Comparison | Total n | Silenced n | p-value | Strains | Notes |
|--------|-----------|---------|-----------|---------|---------|-------|
| 6A | 8 days post RNAi vs 7 days on RNAi | 124, 55 | 45, 51 | <0.00001, * | N2 | |
| S2B | wild type vs rde-10(jam243) | 115, 102 | 115, 100 | 0.13140071, ns | N2, AMJ1614 | |
| S2B | wild type vs mut-16(jam240) rde-10(jam206) | 115, 137 | 115, 5 | <0.00001, * | N2, AMJ1611 | |
| S2C | wild type vs mut-16(jam268) | 322, 74 | 279, 20 | <0.00001, * | N2, AMJ1675 | From the same feed in *Figure 2E* |
| S6B | wild type unc-22 vs wild type pL4440 | 138, 116 | 138, 0 | <0.00001, * | N2 | |
| S6B | wild type unc-22 vs nrde-3(jam205) unc-22 | 138, 110 | 138, 79 | <0.00001, * | N2, AMJ1510 | |
| S6B | unc-22(jam300) unc-22 vs unc-22(jam300) pL4440 | 101, 113 | 101, 101 | 0.00128, * | AMJ1730 | |
| S6B | unc-22(jam301) unc-22 vs unc-22(jam301) pL4440 | 101, 65 | 101, 49 | <0.00001, * | AMJ1731 | |
| S6B | unc-22(jam300); nrde-3(jam205) unc-22 vs unc-22(jam300); nrde-3(jam205) pL4440 | 82, 108 | 70, 83 | 0.14214, ns | AMJ1757 | |
| S6B | unc-22(jam300); nrde-3(jam205) unc-22 vs unc-22(jam300) unc-22 | 82, 101 | 70, 101 | <0.00001, * | AMJ1757, AMJ1730 | |
| S6B | unc-22(jam300); nrde-3(jam205) unc-22 vs nrde-3(jam205) unc-22 | 82,110 | 70, 79 | 0.02592, * | AMJ1757, AMJ1510 | |

The online version of this article includes the following source data for table 3:

**Source data 1.** Excel sheet containing all information in *Table 3*.

of binding by different small RNAs to the same intact mRNA or pre-mRNA molecules is not allowed, and (4) there are no other mechanisms for the turnover of the RNA species considered in the timescale considered. Specific details areprovided within the script '2022_2_9_RNAi_network_thresholds_ simpler.R' available on GitHub (copy archived at *Knudsen et al., 2024*).

## Dynamic model

A series of differential equations were used to describe the rate of change for dsRNA, 1° siRNAs, mRNAs, pre-mRNAs, pUG RNAs, and 2° siRNAs, and numerically simulated using the 4th Order Runge Kutta method. Specific details are provided within the scripts '2022_6_29_Celegans_RNAi_ ODEs_RK4_method_d6.py' and '2022_7_14_RNAiDynamics_ODEs_Parameter_Analysis.R' available on GitHub (copy archived at *Knudsen et al., 2024*).

## Genome editing

The gonads of adult *C. elegans* were injected with nuclear-localized Cas9 (PNA Bio) preincubated at 37°C for 10 min with a hybridized crRNA/tracrRNA (Integrated DNA Technologies), as well as an oligonucleotide or PCR-amplified homology repair template. Plates with successfully edited F1 animals were identified by screening the Dpy or Rol animals obtained when using *dpy-10* editing as a co-CRISPR (*Arribere et al., 2014*; *Paix et al., 2015*) or for Rol animals when using the pRF4 plasmid as a co-injection marker (*Dokshin et al., 2018*).

To introduce a premature stop codon in *mut-16*: Injection of a crRNA with the target sequence (P9) (Integrated DNA Technologies), tracrRNA, Cas9, a *mut-16(−)* homology repair template (P10) mimicking the mutation in *mut-16(jam139)*, predicted amino acid change Y294*, and *dpy-10* crRNA (P11) and *dpy-10(−)* homology repair template (P12) into N2 or AMJ1489 and subsequent screening were performed as described above. Genotyping for *mut-16(jam148, jam240, jam265, jam266, jam267, or jam268)* was performed using duplex PCR (P13, P14) followed by restriction digestion

with BstBI. The nonsense mutations in different strains (AMJ1397, AMJ1611, AMJ1672, AMJ1673, AMJ1674, and AMJ1675) were verified by Sanger sequencing.

To make the *mut-16(−) rde-10(−)* double mutant: Injection of a crRNA with the target sequence (P15) (Integrated DNA Technologies), tracrRNA, Cas9, a *rde-10(−)* homology repair template (P16) mimicking the mutation in *rde-10(jam248)*, and *dpy-10* crRNA (P11) and *dpy-10(−)* homology repair template (P12) into AMJ1397 (*mut-16(jam148)*) and subsequent screening were performed as described above. Genotyping for *rde-10(−)* was performed using duplex PCR (P17, P18) followed by restriction digestion with EcoRV. A strain with a mutation in *rde-10* that results in a 115-bp frameshift followed by an early stop codon was designated as AMJ1470.

To introduce the mutation in *rde-10* that will encode RDE-10(Ser228Phe): Injection of a crRNA with the target sequence (P15) (Integrated DNA Technologies), tracrRNA, Cas9, a *rde-10(−)* homology repair template (P16) mimicking the mutation in *rde-10(jam248)* (**Figure 1—figure supplement 1**), and pRF4 into N2 and subsequent screening were performed as described above. Genotyping for the mutation was performed using duplex PCR (P17, P18) followed by restriction digestion with EcoRV. A strain with the missense mutation verified by Sanger sequencing was designated as AMJ1489.

To introduce a premature stop codon in *rde-10*: Injection of a crRNA with the target sequence listed as (P19) (Integrated DNA Technologies), tracrRNA, Cas9, a *rde-10(−)* homology repair template (P20) (predicted amino acid change Q73*) and pRF4 into N2 and subsequent screening were performed as described above. Genotyping for *rde-10(−)* was performed using duplex PCR (P21, P22) and restriction digestion with DpnII to isolate the mutant from N2. A strain with a 2-bp deletion near Q73 that results in a frameshift and an early stop codon was designated as AMJ1614.

To introduce a premature stop codon in *nrde-3*: Injection of a crRNA with the target sequence (P23) (Integrated DNA Technologies), tracrRNA, Cas9, a *nrde-3(−)* homology repair template (P24), mimicking *nrde-3(gg066)* (25), and pRF4 into N2 and subsequent screening were performed as described above. Genotyping for *nrde-3(jam205)* was performed using duplex PCR (P25, P26) followed by restriction digestion with AclI. A strain with the nonsense mutation verified by Sanger sequencing was designated as AMJ1510.

To introduce a premature stop codon in *eri-1*: Injection of a crRNA with the target sequence (P27) (Integrated DNA Technologies), tracrRNA, Cas9, an *eri-1(−)* homology repair template (P28), predicted to encode ERI-1(E225*) after the edit, and pRF4 into AMJ1611 or N2 and subsequent screening were performed as described above. Genotyping for *eri-1(jam260, jam261, jam262, jam263, or jam264)* was performed using duplex PCR (P29, P30) followed by restriction digestion with DpnII. Additionally, when *eri-1(mg366)* was crossed with other mutants, duplex PCR with P31 and P32 was used for genotyping.

To create a transgene with the *bli-1* promoter: Injection of two crRNAs with the target sequences (P33, P34) (Integrated DNA Technologies), tracrRNA, Cas9, a homology repair template that was amplified using sequences (P35, P36) (Phusion High-Fidelity DNA Polymerase, New England BioLabs), which amplifies the promoter region of *bli-1*, and pRF4 into AMJ1195 [*mex-5p::gfp::cye-1 3′ utr*] (40) and subsequent screening were performed as described above. Genotyping for *bli-1p::gfp* was performed using triplex PCR (P37, P38, P39). Additional genotyping after crosses was done using triplex PCR with sequences (P40, P41, P42). The resulting strain (AMJ1709) resulted in successful integration of ~75% of the *bli-1* promoter upstream of the *mex-5* promoter, and showed GFP expression within the hypodermis (most notable in the head region, see **Figure 5C**) and in the germline (data not shown).

To mutate the 3′*cis*-regulatory regions of *bli-1*: Injection of two crRNAs with the target sequences (P43, P44) (Integrated DNA Technologies), tracrRNA, Cas9, a homology repair template that was amplified using sequences (P45, P46) (Phusion High-Fidelity DNA Polymerase, New England BioLabs), which amplifies the 3′utr + 50 bp of *unc-22*, and pRF4 into N2 and subsequent screening were performed as described above. Genotyping for the altered *bli-1* gene was performed using triplex PCR (P47, P48, P49). A strain with partial (~65%) integration of the *unc-22 3′cis* region, a repeat of the first 183 bp of the *unc-22 3′cis* region, and the endogenous *bli-1 3′cis* region was designated as AMJ1754 and a strain with complete integration was designated as AMJ1755.

To mutate the 3′*cis*-regulatory regions of *unc-22*: Injection of two crRNAs with the target sequences (P50, P51) (Integrated DNA Technologies), tracrRNA, Cas9, a homology repair template that was amplified using sequences (P52, P53) (Phusion High-Fidelity DNA Polymerase, New England BioLabs),

which amplifies the 3'utr + 50 bp of *bli-1*, and pRF4 into N2 and subsequent screening were performed as described above. Genotyping for the altered *unc-22* gene was performed using triplex PCR (P47, P48, P54). Strains with mutated *unc-22* 3' *cis* region were isolated and designated as AMJ1730 and AMJ1731.

## Sequence and structure alignments

Sequences of *C. elegans* proteins were obtained from WormBase; sequences of proteins from all other species were obtained from UniProt. Alignments were created using Clustal Omega (EMBL-EBI) with default settings.

PyMOL (v. 2.4.1) was used to modify and annotate PDB files. The RDE-10 (UniProt: Q9N3S2) PDB file is based on predictions from AlphaFold. Protein domains were colored based on homology to domains as found in the EMBL-EBI Pfam database (Maelstrom: PF13017). The protein structure alignment was done using the Pairwise Structure Alignment from Protein Data Bank with rigid parameters (Root mean square deviation (RMSD) cutoff: 3; Aligned fragment pair (AFP) distance cutoff: 1600; fragment length: 8). The exonuclease domain of ERI-1 (UniProt:O444606) and of MUT-7 (UniProt:P34607) was compared with the Maelstrom domain of RDE-10.

## Feeding RNAi

Control RNAi by feeding *E. coli* containing the empty dsRNA-expression vector (L4440), which can generate a short dsRNA derived from the multiple cloning site but does not produce dsRNA against any *C. elegans* gene, was done in parallel with all RNAi assays.

P0 and F1 feeding: Bacteria expressing dsRNA was cultured in LB media with 100 µg/µl carbenicillin overnight at 37°C at 250 rpm. 100 µl of cultured bacteria was then seeded onto RNAi plates [NG agar plate supplemented with 1 mM isopropyl β-D-1-thiogalactopyranoside (Omega) and 25 µg/ml carbenicillin (MP Biochemicals)]. Adult animals were passaged onto seeded RNAi plates and removed after 24 hr. For the weaker RNAi assay described in *Figure 2E*, RNAi plates that were kept at 4°C for over 4 months were seeded. Progeny were scored for silencing by bacteria expressing dsRNA targeting *unc-22* (defect evident as twitching within ~3 min in 3 mM levamisole) or *bli-1* (defect evident as blisters along the body).

P0 pulse feeding: L4 and young adult animals were placed on seeded RNAi plates for 1 hr after which they were transferred to an OP50 plate for 1 hr, and then transferred to a new OP50 plate once again to minimize the residual RNAi food carryover. Animals were left on OP50 plates and scored every 24 hr for 8 subsequent days with transfer to new OP50 plates every 2 days to prevent overcrowding.

F1 only feeding: A single L4 or young adult (1 day older than L4) animal (P0) was placed on an RNAi plate seeded with 5 µl of OP50 and allowed to lay eggs. After 1 day, when most of the OP50 was eaten, the P0 animal was removed, leaving the F1 progeny. 100 µl of an overnight culture of RNAi food (*E. coli* which express dsRNA against a target gene) was added to the plate. Two or three days later, the F1 animals were scored for gene silencing by measuring gene-specific defects.

## RNA extraction and PCR with reverse transcription

Total RNA was extracted using TRIzol (Fisher Scientific) from pellets of mixed-stage animals collected from non-starved but crowded plates in biological triplicate for each strain after exposure to either *unc-22* RNAi or the L4440 vector. The aqueous phase was then washed with an equal amount of chloroform and precipitated overnight at −20°C with 10 µg glycogen (Invitrogen) and 1 ml of isopropanol. RNA pellets were washed twice with 70% ethanol and resuspended in 25 µl of nuclease free water.

RT-PCRs for pUG RNAs (*Figure 6C*) were done as described earlier (*Shukla et al., 2020*). Briefly, cDNA was created from isolated RNA using Superscript III Reverse Transcriptase (Thermo Fisher) and a universal primer (P55) that contains nine CA repeats and two adapter sequences. The cDNA was used as a template for a Phusion (New England Biolabs) PCR with the first set of primers (adapter 1 P56; *gsa-1* P57; 0 kb P58; 1 kb P59). The amplicon was diluted 1:20 and used as template for the nested Phusion PCR with the second set of primers (adapter 2 P60; *gsa-1* P61; 0 kb P62; 1 kb P63). The final PCR product (20 µl) was loaded on a 1% agarose gel and imaged. An annealing temperature of 60°C was used for *gsa-1* and 57°C was used for all other primer sets.

For semi-quantitative RT-PCR (*Figure 3—figure supplement 1*), RNA from each strain was isolated from 50 L4-staged animals as described earlier (*Jobson et al., 2015*). Primer P64 was used to reverse transcribe the sense strand of *rde-4* and P65 was used to reverse transcribe the sense strand of *tbb-2*. The resulting cDNA was used as a template for PCR (30 cycles for both *rde-4* and *tbb-2*) using Taq polymerase and gene-specific primers (P66, P67 for *rde-4* and P68, P69 for *tbb-2*). Intensities of the bands were quantified using ImageJ (NIH). The relative intensity of the *rde-4* band normalized to that of the *tbb-2* band was set as 1.0 in wild type. The relative normalized intensity of the *rde-4* band in WM49 (*rde-4(ne301)*) was subtracted from that in AMJ565 to report the levels of *rde-4(+)* mRNA (0.3 relative to wild type).

## Microscopy

Following *gfp* RNAi (*Figure 5C and D*), F1 adult animals were mounted in 10 µl of 3 mM levamisole on a 2% agar pad and imaged under a coverslip using a Nikon AZ100 microscope and Prime BSI Express sCMOS camera. A C-HGFI Intensilight Hg Illuminator was used to excite GFP (filter cube: 450–490 nm excitation, 495 dichroic, and 500–550 nm emission). Representative images for GFP expression were adjusted to identical levels in Fiji (NIH) for presentation.

## Rationale for inferences

### Prior knowledge

Gene-specific requirements for RNA silencing could reflect specialization along pathways, as is supposed for multiple endogenous small RNA pathways in *C. elegans*. Reasons that impact the efficiency of silencing a gene are obscure because of the lack of a quantitative model for RNAi that incorporates recently discovered RNA intermediates.

### Evidence supporting key conclusions

Three different proteins, MUT-16, RDE-10, and NRDE-3, play a role in RNAi such that each is singly required for silencing *bli-1* but any two are sufficient for silencing *unc-22*. Simulations support the parsimonious hypothesis that this difference in requirements can be explained by quantitative contributions from regulators within an intersecting network for silencing both genes but not by parallel pathways downstream of mRNA recognition. Consistently, the requirement for NRDE-3 for silencing *bli-1* is bypassed by enhancing the processing of dsRNA through the loss of ERI-1 or the overexpression of RDE-4.

A quantitative model for RNAi of any gene at steady state reveals several ways that differences in genetic requirements could arise for silencing different genes. Experimental tests that confirm predictions of the quantitative models include changes in the requirement for NRDE-3 for silencing caused by altering *cis*-regulatory regions of the gene targeted by dsRNA; recovery from silencing in non-dividing cells after exposure to a pulse of *unc-22* dsRNA, which supports the turnover of all key RNA intermediates (1° siRNAs, 2° siRNAs, and pUG RNAs) through mechanisms that are currently unknown; and the dearth of pUG RNA generation by 2° siRNAs, consistent with a lack of 3° siRNAs.

## Materials availability

All *C. elegans* strains generated are available upon request.

## Acknowledgements

We thank Mary Chey and Julianna Gross for some analysis of mutants from the genetic screen; Zhongchi Liu, Leslie Pick, Norma Andrews, and members of the Jose Lab for comments on the manuscript; and the *Caenorhabditis elegans* Genetic Stock Center for some worm strains. Funding: This work was supported in part by National Institutes of Health Grants R01GM111457 and R01GM124356, and National Science Foundation Grant 2120895 to AMJ.

## Additional information

### Funding

| Funder | Grant reference number | Author |
|---|---|---|
| National Institutes of Health | R01GM111457 | Antony M Jose |
| National Institutes of Health | R01GM124356 | Antony M Jose |
| National Science Foundation | 2120895 | Antony M Jose |

The funders had no role in study design, data collection, and interpretation, or the decision to submit the work for publication.

### Author contributions

Daphne R Knudsen-Palmer, Conceptualization, Software, Formal analysis, Validation, Investigation, Visualization, Methodology, Writing – original draft, Writing – review and editing, A.M.J. wrote scripts for quantitative modeling that were confirmed and adapted to R by D.R.K; Pravrutha Raman, Farida Ettefa, Laura De Ravin, Formal analysis, Investigation, Writing – review and editing; Antony M Jose, Conceptualization, Resources, Data curation, Software, Formal analysis, Supervision, Funding acquisition, Validation, Investigation, Visualization, Methodology, Writing – original draft, Project administration, Writing – review and editing

### Author ORCIDs

Daphne R Knudsen-Palmer ⓘ http://orcid.org/0000-0001-5745-1185
Antony M Jose ⓘ https://orcid.org/0000-0003-1405-0618

Reviewer #1 (Public Review): https://doi.org/10.7554/eLife.97487.3.sa1
Reviewer #2 (Public Review): https://doi.org/10.7554/eLife.97487.3.sa2
Author response https://doi.org/10.7554/eLife.97487.3.sa3

## Additional files

### Supplementary files

• MDAR checklist

### Data availability

Fastq files from whole-genome sequencing are available at NCBI SRA database with the accession number PRJNA928750. All source data are available at figshare. All code is available on GitHub (copy archived at *Knudsen et al., 2024*).

The following datasets were generated:

| Author(s) | Year | Dataset title | Dataset URL | Database and Identifier |
|---|---|---|---|---|
| Knudsen-Palmer DR, Raman K, Ettefa F, De Ravin L, Jose AM | 2024 | Target-specific requirements for RNA interference can be explained by a single regulatory network | https://ncbi.nlm.nih.gov/bioproject/PRJNA928750 | NCBI Sequence Read Archive, PRJNA928750 |
| Knudsen-Palmer D, Raman P, Ettefa F, De Ravin L, Jose AM | 2024 | Knudsen-Palmer_et_al_2024: Source Data | https://doi.org/10.6084/m9.figshare.24992775.v1 | figshare, 10.6084/m9.figshare.24992775.v1 |

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
