## [Editor Report · eLife assessment]

This **valuable** study shows how an intersecting network of regulators acting on genes with differences in their RNA metabolism explains why the loss of some regulators of RNAi in *C. elegans* can selectively impair the silencing of some target genes. The evidence presented is **convincing**, as the authors use a combination of computational modeling and RNAi assays to support their conclusions.

---

## [Referee Report · Reviewer #1 (Public Review)]

Summary:

The goal of Knudsen-Palmer et al. was to define a biological set of rules that dictate the differential RNAi-mediated silencing of distinct target genes, motivated by facilitating the long-term development of effective RNAi-based drugs/therapeutics. This work provides insights into how (1) cis-regulatory elements influence the RNAi-mediated regulation of genes; (2) determines that genes can "recover" from RNAi-silencing signals in an animal; and (3) pUGylation occurs exclusively downstream of the dsRNA trigger sequence, suggesting 3º siRNAs are not produced. In addition, the authors show that the speed at which RNAi-silencing is triggered does not correlate with the longevity of the silencing. Overall, the work presented supports the conclusions of the authors. The insights are significant because they suggest that if we understand the rules by which RNAi pathways effectively silence genes with different transcription/processing levels then we can design more effective synthetic RNAi-based therapeutics targeting endogenous genes.

Major strength:

The authors use a combination of computational modeling, genetics, and RNAi function assays to reveal several criteria for effective RNAi-mediated silencing of two distinct targets.

Weakness:

It may be beyond the scope of this study, but it would be interesting to know the typical expression levels and turnover rates of unc-22 and bli-1. Based on the results from the altered cis-regulatory regions of bli-1 and unc-22 in Fig 5, it seems like the transcription/turnover rates of each of these genes could also be used as a proof of principle for testing the model proposed in Figure 4. The strength of the model would be further increased if the RNAi sensitivity of unc-22 reflects differences in its transcription/turnover rates compared to bli-1.

---

## [Referee Report · Reviewer #2 (Public Review)]

Summary:

This manuscript by Knudsen-Palmer et al. describes and models the contribution of MUT-16 and RDE-10 in the silencing through RNAi by the Argonaute protein NRDE-3 or others. The authors show that MUT-16 and RDE-10 constitute an intersecting network that can be redundant or not depending on the gene being targeted by RNAi. In addition, the authors provide evidence that increasing dsRNA processing can compensate for NRDE-3 mutants. Overall, the authors provide convincing evidence to understand the factors involved in RNAi in *C. elegans* by using a genetic approach.

Major strengths:

The author's work presents a compelling case for understanding the intricacies of RNA interference (RNAi) within the model organism *Caenorhabditis elegans* through a meticulous genetic approach. By harnessing genetic manipulation, they delve into the role of MUT-16 and RDE-10 in RNAi, offering a nuanced understanding of the molecular mechanisms at play in two independent case study targets (unc-22 and bli-1).

Major weaknesses:

(1) It is unclear how the molecular mechanisms of amplification are different under the MUT-16 and RDE-10 branches of the regulatory pathway, since they are clearly distinct proteins structurally. It would be interesting to do some small-RNA-seq of products generated from unc-22 and bli-1, on wild-type conditions and some of the mutants studied (eg. mut-16, rde-10 and mut-16 + rde-10). That would provide some insights on whether the products of the 2 amplifications are the same in all conditions, just changing in abundance, or whether they are distinct in sequence patterns.

(2) In the same line, Figure 5 aims to provide insights to the sequence determinants that influence on the RNAi of bli-1. It is unclear whether the changes in transcript stability dictated by the 3'UTR are the sole factor governing the preference for the MUT-16 and RDE-10 branches of the regulatory pathway. In line with the mutant jam297, it might be interesting to test whether factors like codon optimality, splicing, ... of the ORF region upstream from bli-1-dsRNA can affect its sensitivity to the MUT-16 and RDE-10 branches of the regulatory pathway.

---

## [Author Response]

The following is the authors’ response to the original reviews

**Public Reviews:**

**Reviewer #1 (Public Review):**
The goal of Knudsen-Palmer et al. was to define a biological set of rules that dictate the differential RNAi-mediated silencing of distinct target genes, motivated by facilitating the long-term development of effective RNAi-based drugs/therapeutics. To achieve this, the authors use a combination of computational modeling and RNAi function assays to reveal several criteria for effective RNAi-mediated silencing. This work provides insights into how (1) cis-regulatory elements influence the RNAi-mediated regulation of genes; (2) it is determined that genes can "recover" from RNAi-silencing signals in an animal; and (3) pUGylation occurs exclusively downstream of the dsRNA trigger sequence, suggesting 3º siRNAs are not produced. In addition, the authors show that the speed at which RNAi-silencing is triggered does not correlate with the longevity of the silencing. These insights are significant because they suggest that if we understand the rules by which RNAi pathways effectively silence genes with different transcription/processing levels then we can design more effective synthetic RNAi-basedtherapeutics targeting endogenous genes. The conclusions of this study are mostly supported by the data, but there are some aspects that need to be clarified.

We thank the reviewer for their kind words and for appreciating the practical utility of our approach and discoveries.

(1) The methods do not describe the "aged RNAi plates feeding assay" in Figure 2E. The figure legend states that "aged RNAi plates" were used to trigger weaker RNAi, but the detail explaining the experiment is insufficient. How aged is aged? If the goal was to effectively reduce the dsRNA load available to the animals, why not quantitatively titrate the dsRNA provided? Were worms previously fed on the plates, or was simply a lawn of bacteria grown until presumably the IPTG on the plate was exhausted?

We have elaborated our methods section to describe that the plates were left at 4ºC for about 4 months before adding bacteria and performing the assay, with one possible reason for the weaker knockdown being that perhaps the IPTG in the RNAi plates is less effective. However, it is worth noting that the robustness of a feeding RNAi assay can vary from culture to culture and/or batch of plates. We therefore always perform RNAi assays with wild-type animals alongside test strains to gauge the strength of the RNAi assay for a given culture and batch of plates. We called the data in Figure 2E “weak” because of the response of wild-type animals was weak as evidenced by weak twitching in levamisole. Despite this reduced effect, we observed 100% penetrance in wild-type animals, enabling us to sensitively detect the reduced responses of the mutants.

(2) Is the data presented in Figure 2F completed using the "aged RNAi plates" to achieve the partial silencing of dpy-7 observed? Clarification of this point would be helpful.

No. The only occasion when plates were older was as in response to comment 1 above.

(3) Throughout the manuscript the authors refer to "non-dividing cells" when discussing animals' ability to recover from RNA silencing. It is not clear what the authors specifically mean with the phrase "non-dividing cells", but as this is referred to in one of their major findings, it should be clarified. Do they mean the cells are somatic cells in aged animals, thus if they are "non-dividing" the siRNA pools within the cells cannot be diluted by cell division? Based on the methods, the animals of RNAi assays were L4/Young adults that were scored over 8 days after the initial pulse of dsRNA feeding. If this is the case, wouldn't these animals be growing into gravid adults after the feeding, and thus have dividing cells as they grew?

We thank the reviewer for highlighting the need to explain this point further. Our experiment test the silencing of the unc-22 gene, which is expressed and functions in body-wall muscle cells. Most of the body wall muscles in *C. elegans* are developed by the L1 stage (reviewed in Krause and Liu, 2012), and they do not divide between the L4 and adult stages. Therefore, during the duration of the experiment where we delivered a pulse of dsRNA and examined responses over days, none of these cells divide. We have added a statement in the main text to explicitly say that the recovery from silencing by dsRNA that we observed cannot be explained by dilution during cell divisions.

(4) What are the typical expression levels/turnover of unc-22 and bli-1? Based on the results from the altered cis-regulatory regions of bli-1 and unc-22 in Figure 5, it seems like the transcription/turnover rates of each of these genes could also be used as a proof of principle for testing the model proposed in Figure 4. The strength of the model would be further increased if the RNAi sensitivity of unc-22 reflects differences in its transcription/turnover rates compared to bli-1.

We can get a sense of the relative abundances of *unc-22* and *bli-1* across development from the RNA-seq experiments that have been performed by others in the field (see below). However, these data cannot be used to infer either the production or the turnover rates. Future experiments that measure production (the combined rate of transcriptional run-on, splicing, export from the nucleus, etc.) will be required to define the production rates. Similarly, assays that detect the rate of degradation of transcripts without confounding presence from continued production will be needed to establish turnover rates. Future efforts to obtain values for these in vivo rates for multiple genes will help further test the model.

**Author response image 1. sa3fig1:** Expression data for unc-22.

**Author response image 2. sa3fig2:** Expression data for bli-1.

**Reviewer #2 (Public Review):**
Summary:This manuscript by Knudsen-Palmer et al. describes and models the contribution of MUT-16 and RDE-10 in the silencing through RNAi by the Argonaute protein NRDE-3 or others. The authors show that MUT-16 and RDE-10 constitute an intersecting network that can be redundant or not depending on the gene being targeted by RNAi. In addition, the authors provide evidence that increasing dsRNA processing can compensate for NRDE-3 mutants. Overall, the authors provide convincing evidence to understand the factors involved in RNAi in *C. elegans* by using a genetic approach.Major Strengths:The author's work presents a compelling case for understanding the intricacies of RNA interference (RNAi) within the model organism *Caenorhabditis elegans* through a meticulous genetic approach. By harnessing genetic manipulation, they delve into the role of MUT-16 and RDE-10 in RNAi, offering a nuanced understanding of the molecular mechanisms at play in two independent case study targets (unc-22 and bli-1).

We thank the reviewer for their kind words and for appreciating our genetic analysis.

Major Weaknesses:(1) It is unclear how the molecular mechanisms of amplification are different under the MUT-16 and RDE-10 branches of the regulatory pathway, since they are clearly distinct proteins structurally. It would be interesting to do some small-RNA-seq of products generated from unc-22 and bli-1, on wild-type conditions and some of the mutants studied (eg. mut-16, rde-10 and mut16 + rde-10). That would provide some insights into whether the products of the 2 amplifications are the same in all conditions, just changing in abundance, or whether they are distinct in sequence patterns.

As we highlight in the paper, MUT-16 and RDE-10 are indeed very different proteins. One possible hypothesis suggested by this difference is that different kinds of small RNAs are made when the underlying mechanism relies on MUT-16 versus on RDE-10. However, postulating such a difference is not necessary for explaining the data. Furthermore, since the amounts of 2º siRNAs do not have to be correlated with the strength of silencing (Figure 4E), this work raises caution against the over-reliance on small RNA sequencing for inferring gene silencing. Nevertheless, it is indeed an attractive possibility that the amounts of small RNA, their distributions along mRNA sequence, and/or the sequence biases of the accumulating small RNAs could be different when relying on MUT-16- or RDE-10-dependent mechanisms. Future work that directly examine the small RNAs that accumulate in different mutant strains after initiating RNAi can shed light on these possibilities.

(2) In the same line, Figure 5 aims to provide insights into the sequence determinants that influence the RNAi of bli-1. It is unclear whether the changes in transcript stability dictated by the 3'UTR are the sole factor governing the preference for the MUT-16 and RDE-10 branches of the regulatory pathway. In line with the mutant jam297, it might be interesting to test whether factors like codon optimality, splicing, ... of the ORF region upstream from bli-1-dsRNA can affect its sensitivity to the MUT-16 and RDE-10 branches of the regulatory pathway.

In Figure 5, we eliminated the possibility that any gene that is transcribed using the *bli-1* promoter would require NRDE-3, and showed using *jam297* that modifications to the 3’ *cis* regulatory regions of a target can alter the dependence on NRDE-3 for knockdown. We agree that future experiments that control individual aspects of *bli-1*, potentially one feature at a time, can reveal the separate contributions of each characteristic of the gene to the observed dependence on NRDE-3 of the wild-type *bli-1* gene. However, given the many ways that the same level of transcript knockdown can be achieved in our modeling (Figure 4 and its supplemental figures) we expect that multiple characteristics could contribute to NRDE-3 dependence.

**Recommendations For The Authors:**

**Reviewer #1 (Recommendations For The Authors):**
(1) On page 5, the authors state that "MUT-16 and RDE-10 are redundantly or additively required for silencing unc-22"; however, based on their data in Figure 1D, it seems nearly 100% silencing of unc-22 is achieved in single mut-16 or rde-10 mutants. If this is the case, wouldn't it suggest that redundancy of MUT-16 and RDE-10, and not an "additive effect" of MUT-16 and RDE-10 function? Although, as the mutator complex nucleates around MUT-16, the data in Figure 1D suggests it is possible that the presence of MUT-16 or RDE-10 is sufficient for the recruitment of one or more factors that triggers the silencing of unc-22, and thus only one of these factors is necessary.

Because we are seeing 100% silencing in wild-type, *mut-16(-)*, or *rde-10(-)* animals in Figure 1D, this assay (where the silencing response is strong) does not allow us to discriminate between differing levels of silencing. The “weak” RNAi assay in Figure 2E provides the opportunity to observe differences in the contributions made by MUT-16 or RDE-10, supporting the idea that the 2º siRNAs and relative contributions to silencing can indeed be additive, explaining the complete loss of silencing only in the double mutant. While MUT-16 has been shown to be required for the recruitment of other *Mutators* in the germline, *Mutator* foci are not detectable in the soma. Given that *unc-22* and *bli-1* are somatic targets, we are hesitant to assume a mechanism for the production of small RNAs that requires a similar MUT-16-dependent nucleation in somatic cells. MUT-16 is clearly required for full silencing. But, if it functions similarly in the soma and the germline remains an open question. Indeed the mechanism(s) for producing small RNAs in somatic cells could be different from that used for production of small RNAs in the germline because of known differences in the use of RNA-dependent RNA polymerases (e.g. Ravikumar et al., Nucleic Acids Res. 2019). Future studies that determine the subcellular localization(s) and potential biochemical function(s) of RDE-10 and MUT-16 in somatic cells are needed to further delineate mechanisms.

(2) On page 10, "rather than one that looks a frequency" - the "a" should be "at".

We thank the reviewer and have fixed this typo.

(3) Figure 4 is very crowded, further dividing 4A (right) and 4B into subpanels would help the readability of the figure.

We thank the reviewer for identifying these figures as being particularly crowded. These panels are presented as single units because the left and right portions of each panel are intimately connected. In Fig. 4A, the outline of mechanism deduced on the left is based on experiments at various scales shown on the right. We have now clarified this in the figure legend. In Fig. 4B, the equations on the right define and use the constants depicted on the left and the definitions below apply to both parts. We have now adjusted both figure parts to make these connections clearer.

(4) References to the subpanels of Figure 4 in the text on page 12 are off from the figure and figure legend.For example:"Overall, τkd and tkd were uncorrelated..." refers to 4C when it should refer to 4D. "However, the maximal amount of 2ºsiRNAs..." refers to 4D when it should refer to 4E. "Additionally, an increase in transcription..." refers to 4E when it should refer to 4F."When a fixed amount of dsRNA was exposed..." refers to 4F when it should refer to 4G.

We thank the reviewer for catching these errors and we have corrected these figure references.

**Reviewer #2 (Recommendations For The Authors):**
I would encourage the authors to follow up on some of the more mechanistic comments made above, that would strengthen and complement the genetic part of the work presented.

We agree that additional work is needed to elucidate differences in molecular mechanisms for amplifying small RNAs in an MUT-16-dependent vs. RDE-10-dependent manner. We hope to address these extensions of our work in future manuscripts that focus on the biochemistry of these proteins and the populations of small RNAs generated using them.

I appreciate the efforts to computationally model the dynamics of the system, but I am not sure that it helps that the mathematical modelling treats both branches of the pathway as functionally equals, since they could have some mechanistic specialisation that is not yet elucidated by the current work.

Our assumption that both branches are equivalent is the most parsimonious. If we allowed for differences, even more values for the parameters of the model will agree with experimental data. The strength of the model is that despite such conservative assumptions, it agrees with experimental data. Biochemical elaborations that make the MUT-16 and RDE-10 branches qualitatively different could exist in vivo as suggested by the reviewer. Even with such qualitative differences in detail, the overall impact on gene silencing is a quantitative and additive one as demonstrated by our experiments. Future experimental work focused on biochemistry could elucidate how a Maelstrom domain-containing protein (RDE-10) and an intrinsically disordered protein (MUT-16) act differently to ultimately promote small RNA production.